# GROUP EQUIVARIANT GENERATIVE ADVERSARIAL NETWORKS

**Neel Dey**[*]
New York University
neel.dey@nyu.edu

**Antong Chen & Soheil Ghafurian**[†]
Data Science & Scientific Informatics, Merck & Co., Inc.
antong.chen@merck.com, soheilghafurian@gmail.com

## ABSTRACT

Recent improvements in generative adversarial visual synthesis incorporate real and fake image transformation in a self-supervised setting, leading to increased stability and perceptual fidelity. However, these approaches typically involve image augmentations via additional regularizers in the GAN objective and thus spend valuable network capacity towards approximating transformation equivariance instead of their desired task. In this work, we explicitly incorporate inductive symmetry priors into the network architectures via group-equivariant convolutional networks. Group-convolutions have higher expressive power with fewer samples and lead to better gradient feedback between generator and discriminator. We show that group-equivariance integrates seamlessly with recent techniques for GAN training across regularizers, architectures, and loss functions. We demonstrate the utility of our methods for conditional synthesis by improving generation in the limited data regime across symmetric imaging datasets and even find benefits for natural images with preferred orientation.

## 1 INTRODUCTION

Generative visual modeling is an area of active research, time and again finding diverse and creative applications. A prevailing approach is the generative adversarial network (GAN), wherein density estimation is implicitly approximated by a min-max game between two neural networks (Goodfellow et al., 2014). Recent GANs are capable of high-quality natural image synthesis and scale dramatically with increases in data and compute (Brock et al., 2018). However, GANs are prone to instability due to the difficulty of achieving a local equilibrium between the two networks. Frequent failures include one or both networks diverging or the generator only capturing a few modes of the empirical distribution. Several proposed remedies include modifying training objectives (Arjovsky et al., 2017; Jolicoeur-Martineau, 2018), hierarchical methods (Karras et al., 2017), instance selection (Sinha et al., 2019; 2020), latent optimization (Wu et al., 2019), and strongly regularizing one or both networks (Gulrajani et al., 2017; Miyato et al., 2018; Dieng et al., 2019), among others. In practice, one or all of the above techniques are ultimately adapted to specific use cases.

Further, limits on data quantity empirically exacerbate training stability issues more often due to discriminator overfitting. Recent work on GANs for small sample sizes can be roughly divided into transfer learning approaches (Wang et al., 2018; Noguchi & Harada, 2019; Mo et al., 2020; Zhao et al., 2020a) or methods which transform/augment the available training data and provide the discriminator with auxiliary tasks. For example, Chen et al. (2019) propose a multi-task discriminator which additionally predicts the degree by which an input image has been rotated, whereas Zhang et al. (2020); Zhao et al. (2020c) incorporate consistency regularization where the discriminator is penalized towards similar activations for transformed/augmented real and fake images. However, with consistency regularization and augmentation, network capacity is spent learning equivariance to transformation as opposed to the desired task and equivariance is not guaranteed.

In this work, we consider the problem of training *tabula rasa* on limited data which possess global and even local symmetries. We begin by noting that GANs ubiquitously use convolutional layers

---

[*]Work started and partially done during an internship at Merck & Co., Inc.
[†]Work done while employed at Merck & Co., Inc.

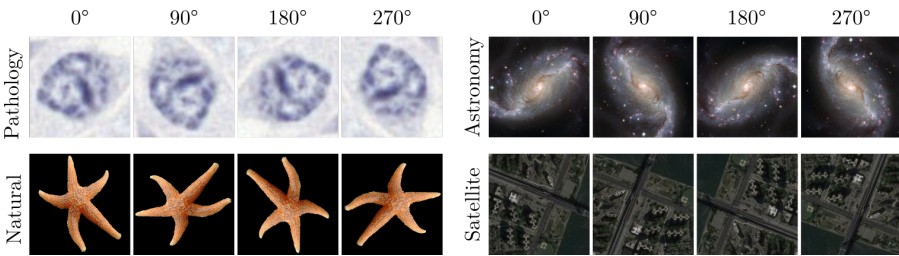

Figure 1: Several image modalities have no preferred orientation for tasks such as classification. We improve their generative modeling by utilizing image symmetries within a GAN framework.

which exploit the approximate translation invariance and equivariance of image labels and distributions, respectively. Equivariance to geometric transformations is key to understanding image representations (Bietti & Mairal, 2019). Unfortunately, other symmetries (e.g., rotations and reflections) inherent to modalities such as astronomy and medical imaging where galaxies and cells can be in arbitrary orientations are not accounted for by standard convolutional layers. To this end, Cohen & Welling (2016) proposed a group-theoretic generalization of convolutional layers (group-convolutions) which in addition to translation, exploit other inherent symmetries and increase the expressive capacity of a network thereby increasing its sample efficiency significantly in detection (Winkels & Cohen, 2019), classification (Veeling et al., 2018), and segmentation (Chidester et al., 2019). Importantly, equivariant networks outperform standard CNNs trained with augmentations from the corresponding group (Veeling et al., 2018, Table 1), (Lafarge et al., 2020a, Fig. 7). See Cohen et al. (2019); Esteves (2020) for a formal treatment of equivariant CNNs.

Equivariant features may also be constructed via scattering networks consisting of non-trainable Wavelet filters, enabling equivariance to diverse symmetries (Mallat, 2012; Bruna & Mallat, 2013; Sifre & Mallat, 2013). Generative scattering networks include Angles & Mallat (2018) where a standard convolutional decoder is optimized to reconstruct images from an embedding generated by a fixed scattering network and Oyallon et al. (2019) who show preliminary results using a standard convolutional GAN to generate scattering coefficients. We note that while both approaches are promising, they currently yield suboptimal synthesis results not comparable to modern GANs. Capsule networks (Hinton et al., 2011; Sabour et al., 2017) are also equivariant and emerging work has shown that using a capsule network for the GAN discriminator (Jaiswal et al., 2019; Upadhyay & Schrater, 2018) improves synthesis on toy datasets. However, capsule GANs and generative scattering approaches require complex training strategies, restrictive architectural choices not compatible with recent insights in GAN training, and have not yet been shown to scale to real-world datasets.

In this work, we improve the generative modeling of images with transformation invariant labels by using an inductive bias of symmetry. We replace all convolutions with group-convolutions thereby admitting a higher degree of weight sharing which enables increased visual fidelity, especially with limited-sample datasets. To our knowledge, we are the first to use group-equivariant layers in the GAN context and to use symmetry-driven considerations in both generator and discriminator architectures. Our contributions are as follows,

1. We introduce symmetry priors via group-equivariance to generative adversarial networks.
2. We show that recent insights in improving GAN training are fully compatible with group-equivariance with careful reformulations.
3. We improve class-conditional image synthesis across a diversity of datasets, architectures, loss functions, and regularizations. These improvements are consistent for both symmetric images and even natural images with preferred orientation.

## 2 METHODS

### 2.1 PRELIMINARIES

**Groups and group-convolutions.** A group is a set with an endowed binary function satisfying the properties of closure, associativity, identity, and invertibility. A two-dimensional symmetry group is

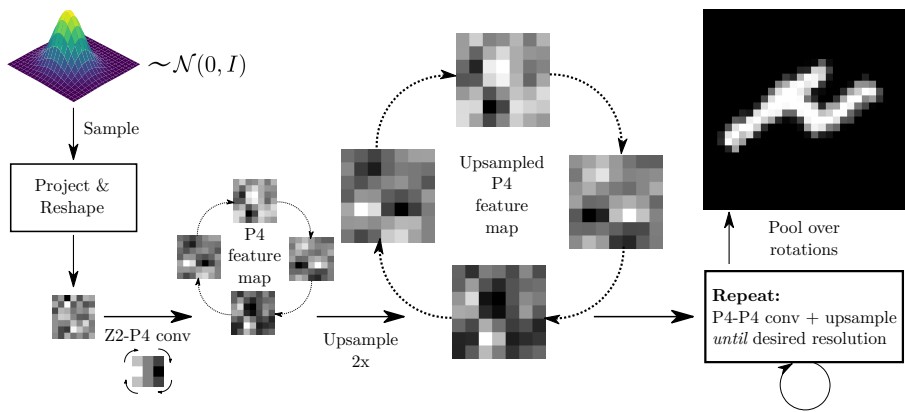

Figure 2: An abbreviated illustration of group-convolutions used in our generator networks.

the set of all transformations under which a geometric object is invariant with an endowed operation of composition. Given a group $G$ and a map $\Phi : X \to Y$ between two $G$-sets $X$ and $Y$, $\Phi$ is said to be *equivariant* i.f.f. $\Phi(g \cdot x) = g \cdot \Phi(x)$, $\forall x \in X$, $\forall g \in G$. Colloquially, an equivariant map implies that transforming an input and applying the map yields the same result as applying the map and then transforming the output. Analogously, invariance requires that $\Phi(g \cdot x) = \Phi(x)$, $\forall x \in X$, $\forall g \in G$. In deep networks, equivariance to a planar symmetry group can be achieved by either transforming filters (Cohen & Welling, 2016) or feature maps (Dieleman et al., 2016).

Our work utilizes the plane symmetry groups $p4$ (all compositions of 90-degree rotations and translations) and $p4m$ (all compositions of 90-degree rotations, reflections, and translations) (Schattschneider, 1978). These groups can be parameterized neatly following Cohen & Welling (2016),

$$g(r, u, v) = \begin{bmatrix} \cos(\frac{r\pi}{2}) & -\sin(\frac{r\pi}{2}) & u \\ \sin(\frac{r\pi}{2}) & \cos(\frac{r\pi}{2}) & v \\ 0 & 0 & 1 \end{bmatrix}; \quad g'(m, r, u, v) = \begin{bmatrix} (-1)^m \cos(\frac{r\pi}{2}) & (-1)^{m+1}\sin(\frac{r\pi}{2}) & u \\ \sin(\frac{r\pi}{2}) & \cos(\frac{r\pi}{2}) & v \\ 0 & 0 & 1 \end{bmatrix}$$

where $g(r, u, v)$ parameterizes $p4$, $g'(m, r, u, v)$ parameterizes $p4m$, $0 \leq r < 4$ (the number of 90-degree rotations), $m \in \{0, 1\}$ (the number of reflections), and $(u, v) \in \mathbb{Z}^2$ (integer translations). The group operation is matrix multiplication for both groups. The matrix $g(r, u, v)$ rotates and translates a point (expressed as homogeneous coordinate vector) in pixel space via left-multiplication. Analogous intuition follows for $g'(m, r, u, v)$.

We now briefly define $G$-equivariant convolutions. We note that formally these are *correlations* and not convolutions and that the literature uses the terms interchangeably. A $G$-convolution between a vector-valued $K$-channel image $f : \mathbb{Z}^2 \to \mathbb{R}^K$ and filter $\psi : \mathbb{Z}^2 \to \mathbb{R}^K$ with $f = (f_1, f_2, \ldots, f_k)$ and $\psi = (\psi_1, \psi_2, \ldots, \psi_k)$ can be expressed as $[f * \psi](g) = \sum_{y \in \mathbb{Z}^2} \sum_{k=1}^{K} f_k(y) \psi_k(g^{-1}y)$. For standard reference, if one considers $G$ to be the translation group on $\mathbb{Z}^2$, we have $g^{-1}y = y - g$ and recover the standard convolution. After the first layer of a $G$-CNN, we see that $(f * \psi)$ is a function on $G$, necessitating that filter banks also be functions on $G$. Subsequent $G$-convolutional layers are therefore defined as $[f * \psi](g) = \sum_{h \in G} \sum_{k=1}^{K} f_k(h) \psi_k(g^{-1}h)$. Finally, for tasks where the output is an image, it is necessary to bring the domain of feature maps from $G$ back to $\mathbb{Z}^2$. We can pool the feature map for each filter over the set of transformations, corresponding to average or max pooling over the group of rotations (or roto-reflections as appropriate).

**GAN optimization and stability.** As we focus on the limited data setting where training instability is exacerbated, we briefly describe the two major stabilizing methods used in all experiments here. We regularize the discriminator by using a zero-centered gradient penalty (GP) on the real data as proposed by Mescheder et al. (2018) of the form, $R_1 := \frac{\gamma}{2} \mathbb{E}_{x \sim \mathbb{P}_{real}}[\|\nabla D(x)\|_2^2]$, where $\gamma$ is the regularization weight, $x$ is sampled from the real distribution $\mathbb{P}_{real}$, and $D$ is the discriminator. This GP has been shown to cause convergence (in toy cases), alleviate catastrophic forgetting (Thanh-Tung & Tran, 2018), and strongly stabilize GAN training. However, empirical work has found that this GP achieves stability at the cost of worsening GAN evaluation scores (Brock et al., 2018).

A widely used technique for GAN stabilization is spectral normalization (Miyato et al., 2018), which constrains the discriminator to be 1-Lipschitz, thereby improving gradient feedback to the generator (Zhou et al., 2019; Chu et al., 2020). With spectral normalization, each layer is rescaled as, $W_{SN} = W/\sigma(W)$, where $W$ is the weight matrix for a given layer and $\sigma(W)$ is its spectral norm. In practice, $\sigma(W)$ is estimated via a power iteration method as opposed to computing the full singular value decomposition during each training iteration. Finally, applying spectral normalization to both generator and discriminator empirically improves training significantly (Zhang et al., 2018).

## 2.2 GROUP EQUIVARIANT GENERATIVE ADVERSARIAL NETWORKS

Here, we outline how to induce a symmetry prior into the GAN framework. Implementations are available at `https://github.com/neel-dey/equivariant-gans`. The literature has developed several techniques for normalization and conditioning of the individual networks, along with unique architectural choices - we extend these developments to the equivariant setting. We start by replacing all convolutional layers with group-convolutional layers where filters and feature maps are functions on a symmetry group $G$. Batch normalization moments (Ioffe & Szegedy, 2015) are calculated per group-feature map as opposed to spatial feature maps. Pointwise nonlinearities preserve equivariance for the groups considered here. Pre-activation residual blocks common to modern GANs are used freely as the sum of equivariant feature maps on $G$ is also equivariant.

**Generator.** The generator is illustrated at a high-level in Figure 2. We use a fully connected layer to linearly project and reshape the concatenated noise vector $z \sim \mathcal{N}(0, I)$ and class embedding $c$ into spatial feature maps on $\mathbb{Z}^2$. We then use spectrally-normalized group-convolutions, interspersed with pointwise-nonlinearities, and nearest-neighbours upsampling to increase spatial extent. We use upsampling followed by group-convolutions instead of transposed group-convolutions to reduce checkerboard artefacts (Odena et al., 2016). We further use a novel group-equivariant class-conditional batch normalization layer (described below) to normalize and class-condition image generation while also projecting the latent vector $z$ to each level of the group-convolutional hierarchy. We finally max-pool over the set of transformations to obtain the generated image $x$.

**Discriminator.** The group-equivariant discriminator receives an input $x$, which it maps to a scalar indicating whether it is real or fake. We do this via spectrally normalized group-convolutions, pointwise-nonlinearities, and spatial-pooling layers to decrease spatial extent. After the final group-convolutional layer, we pool over the group and use global average pooling to obtain an invariant representation at the output. Finally, we condition the discriminator output via the projection method proposed by Miyato & Koyama (2018). Importantly, the equivariance of group-convolutions depends on the convolutional stride. Strided convolutions were commonly used for downsampling in early GANs (Radford et al., 2015). However, stride values must be adjusted to the dataset to preserve equivariance, which makes comparisons to equivalent non-equivariant GAN architectures difficult. We therefore use pooling layers over the plane (commonly used in recent GANs) to downsample in all settings to preserve equivariance and enable a fair comparison.

**Spectral Normalization.** As the singular values of a matrix are invariant under compositions of 90-degree rotations, transpositions, and reflections - spectral normalization on a group-weight matrix preserves equivariance and we use it freely.

**Class-conditional Batch Normalization.** Conditional batch normalization (Perez et al., 2018) replaces the scale and shift of features with an affine transformation learned from the class label (and optionally from the latent vector as well (Brock et al., 2018)) via linear dense layers, and is widely used in generative networks. We propose a group-equivariance preserving conditional normalization by learning the affine transformation parameters per group-feature map, rather than each spatial feature. As we use fewer group-filters than equivalent non-equivariant GANs, we use fewer dense parameters to learn conditional scales and shifts.

## 3 EXPERIMENTS

**Common setups.** In each subsection, we list specific experimental design choices with full details available in App. C. For each comparison, the number of group-filters in each layer is divided by the square root of the cardinality of the symmetry set to ensure a similar number of parameters to the standard CNNs to enable fair comparison. We skew towards stabilizing training over absolute

Table 1: A summary of the datasets considered in this paper. The right-most column indicates whether the dataset has a preferred pose.

| Dataset | Resolution | $n_{\text{classes}}$ | $n_{\text{training}}$ | $n_{\text{validation}}$ | Pose Preference |
|---------|-----------|----------|-----------|-------------|-----------------|
| Rotated MNIST | (28, 28) | 10 | 12,000 | 50,000 | No |
| ANHIR | (128, 128, 3) | 5 | 28,407 | 9,469 | No |
| LYSTO | (256, 256, 3) | 3 | 20,000 | - | No |
| CIFAR-10 | (32, 32, 3) | 10 | 50,000 | 10,000 | Yes |
| Food-101 | (64, 64, 3) | 101 | 75,747 | 25,250 | Yes |

Table 2: Min. & mean Fréchet distances (lower is better) of generated RotMNIST samples, evaluated at every 1K generator iterations. All evaluations are visualized in Appendix A Figure 6.

| Loss | Setting | Min. & Mean Fréchet Distance Available Training Data | | | |
|------|---------|------|------|------|------|
| | | 10% | 33% | 66% | 100% |
| - | Real data | 0.6854 | 0.3208 | 0.1324 | 0.1296 |
| RAGAN | CNN in G & D | (2.04, 11.40) | (1.42, 11.65) | (1.20, 11.10) | (1.36, 11.68) |
| | CNN in G & G-CNN in D | (1.84, 4.26) | (0.88, **3.26**) | (0.52, 2.85) | (0.53, 3.12) |
| | G-CNN in G & CNN in D | (1.49, 9.75) | (1.08, 9.29) | (0.90, 8.70) | (0.95, 9.62) |
| | G-CNN in G & D | (1.61, **4.25**) | (0.76, 3.40) | (0.54, 2.92) | (0.53, **2.90**) |
| NSGAN | CNN in G & D | (**1.00**, 7.02) | (0.74, 8.25) | (0.84, 8.07) | (0.97, 8.49) |
| | CNN in G & G-CNN in D | (2.77, 5.48) | (1.02, 3.51) | (0.55, **2.85**) | (0.54, 3.08) |
| | G-CNN in G & CNN in D | (**1.00**, 7.00) | (0.96, 7.42) | (0.87, 6.83) | (0.94, 7.52) |
| | G-CNN in G & D | (2.85, 5.67) | (1.04, 4.24) | (0.82, 3.27) | (0.64, 3.32) |
| WGAN | CNN in G & D | (3.42, 16.21) | (3.90, 18.32) | (3.87, 17.81) | (4.88, 19.40) |
| | CNN in G & G-CNN in D | (2.87, 5.98) | (0.76, 4.11) | (**0.50**, 3.57) | (**0.39**, 3.51) |
| | G-CNN in G & CNN in D | (2.67, 16.02) | (3.40, 17.03) | (3.77, 17.76) | (3.74, 17.82) |
| | G-CNN in G & D | (2.51, 5.67) | (**0.58**, 3.32) | (0.56, 3.52) | (0.54, 3.76) |

performance to compare models under the same settings to obviate extensive checkpointing typically required for BigGAN-like models. Optimization is performed via Adam (Kingma & Ba, 2014) with $\beta_1 = 0.0$ and $\beta_2 = 0.9$, as in Zhang et al. (2018); Brock et al. (2018). Unless otherwise noted, all discriminators are updated twice per generator update and employ unequal learning rates for the generator and discriminator following Heusel et al. (2017). We use an exponential moving average ($\alpha = 0.9999$) of generator weights across iterations when sampling images as in Brock et al. (2018). All initializations use the same random seed, except for RotMNIST where we average over 3 random seeds. An overview of the small datasets considered here is presented in Table 1.

**Evaluation methodologies.** GANs are commonly evaluated by embedding the real and generated images into the feature space of an ImageNet pre-trained network where similarity scores are computed. The Fréchet Inception Distance (FID) (Heusel et al., 2017) jointly captures sample fidelity and diversity and is presented for all experiments. To further evaluate both aspects explicitly, we present the improved precision and recall scores (Kynkäänniemi et al., 2019) for ablations on real-world datasets. As the medical imaging datasets (ANHIR and LYSTO) are not represented in ImageNet, we finetune Inception-v3 (Szegedy et al., 2016) prior to feature extraction for FID calculation as in Huang et al. (2018). For RotMNIST, we use features derived from the final pooling layer of the $p4$-CNN defined in Cohen & Welling (2016) to replace Inception-featurization. An analogous approach was taken in Binkowski et al. (2018) in their experiments on the canonical MNIST dataset. Natural image datasets (Food-101 and CIFAR-10) are evaluated with the official `Tensorflow` Inception-v3 weights. Importantly, we perform ablation studies on all datasets to evaluate group-equivariance in either or both networks.

We note that the FID estimator is strongly biased (Binkowski et al., 2018) and work around this limitation by always generating the same number of samples as the validation set as recommended in Binkowski et al. (2018). An alternative Kernel Inception Distance (KID) with negligible bias has

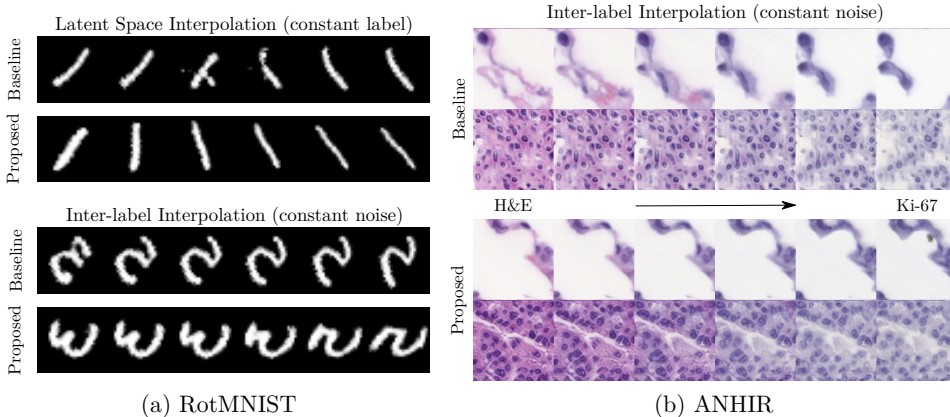

Figure 3: Qualitative GAN interpolation (White, 2016) results. **(a)** Selected spherical interpolations between generated RotMNIST samples in either latent space (**top**) or between labels (**bottom**). Equivariant GANs interpolate intuitively between samples, whereas standard GANs do not. **(b)** Selected inter-label linear interpolations between two staining dyes in synthesized ANHIR images. The standard model (**top**) changes both structure and dye between the generated samples, whereas the equivariant model (**bottom**) better preserves structure while translating between dyes.

been proposed (Binkowski et al., 2018), yet large-scale evaluation (Kurach et al., 2019) finds that KID correlates strongly with FID. We thus focus on FID in our experiments in the main text.

### 3.1 SYNTHETIC EXPERIMENTS: ROTATED MNIST

Rotated MNIST (Larochelle et al., 2007) provides random rotations of the MNIST dataset and is a common benchmark for equivariant CNNs which we use to measure sensitivity to dataset size, loss function, and equivariance in either network to motivate choices for real-world experiments. We experiment with four different proportions of training data: 10%, 33%, 66%, and 100%. Additionally, the non-saturating loss (Goodfellow et al., 2014) (NSGAN), the Wasserstein loss (Arjovsky et al., 2017) (WGAN), and the relativistic average loss (Jolicoeur-Martineau, 2018) (RaGAN) are tested. For the equivariant setting, all convolutions are replaced with $p4$-convolutions. $p4m$ is precluded as some digits do not possess mirror symmetry. All settings were trained for 20,000 generator iterations with a batch size of 64. Implementation details are available in Appendix C.2.1.

**Results.** Fréchet distance of synthesized samples to the validation set is calculated at every thousand generator iterations. As shown in Table 2, we find that under nearly every configuration of loss and data availability considered, using $p4$-convolutions in either network improves both the mean and minimum Fréchet distance. As data availability increases, the best-case minimum and mean FID scores improve. With $\{33\%, 66\%, 100\%\}$ of the data, most improvements come from using a $p4$-discriminator, with the further usage of a $p4$-generator only helping in a few cases. At 10% data, having an equivariant generator is more impactful than an equivariant discriminator. These trends are further evident from App. A Fig. 6, where we see that GANs with $p4$-discriminators converge faster than non-equivariant counterparts. The NSGAN-GP and RAGAN-GP losses perform similarly, with WGAN-GP underperforming initially and ultimately achieving comparable results. Qualitatively, the equivariant model learns better representations as shown in Figure 3(a). Holding the class-label constant and interpolating between samples, we find that the standard GAN changes the shape of the digit in order to rotate it, whereas the equivariant model learns rotation in the latent space. Holding the latent constant and interpolating between classes shows that our model learns an intuitive interpolation between digits, whereas the standard GAN transforms the image immediately.

### 3.2 REAL-WORLD EXPERIMENTS

**Datasets.** $p4$ and $p4m$-equivariant networks are most useful when datasets possess global roto(-reflective) symmetry, yet have also been shown to benefit generic image representation due to local symmetries (Cohen & Welling, 2016; Romero et al., 2020). To this end, we experiment with two

Table 3: FID evaluation (lower is better) of all real-world datasets across ablations and augmentation-based baseline comparisons. - indicates an inapplicable setting for the method.

|  | Setting | ANHIR | LYSTO | CIFAR-10 | Food-101 |
|---|---|---|---|---|---|
| Ablation | CNN in G & D | 7.32 | 7.27 | 20.89 | 27.34 |
|  | G-CNN in G; CNN in D | 6.93 | 6.68 | 21.20 | 24.16 |
|  | CNN in G; G-CNN in D | 5.56 | 5.02 | **17.09** | **16.91** |
|  | G-CNN in G & D | **5.54** | **3.90** | 17.49 | 17.73 |
| + Aug. | CNN in G & D + Standard Aug. | 7.57 | 6.59 | 37.41 | 35.18 |
|  | CNN in G & D + bCR (Zhao et al., 2020c) | 5.86 | 4.78 | 19.64 | 21.18 |
|  | CNN in G & D + AR (Chen et al., 2019) | - | - | 19.59 | 20.39 |
|  | G-CNN in G & D + bCR (Zhao et al., 2020c) | **5.19** | **4.53** | **17.94** | **15.55** |

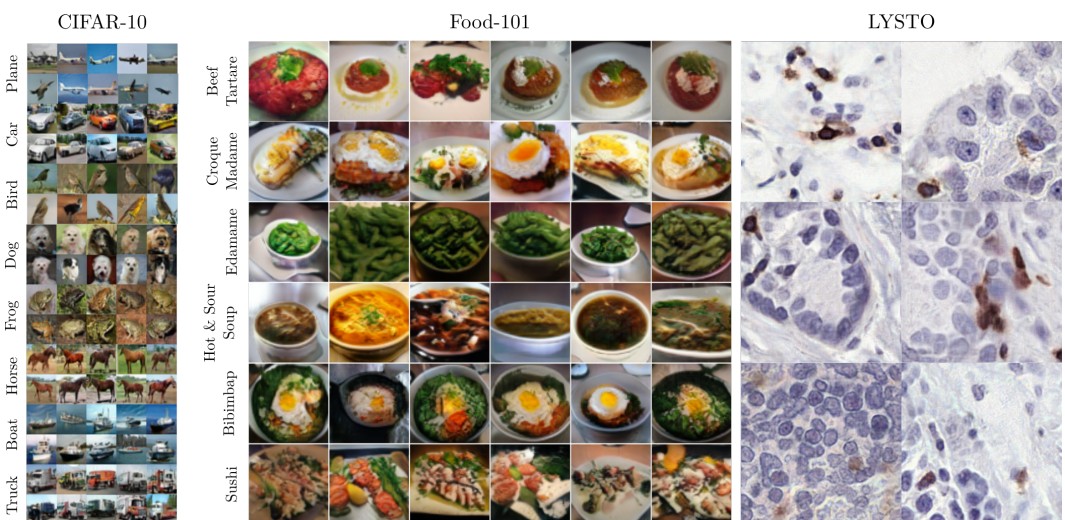

Figure 4: Selected generated samples using the best performing equivariant models with no augmentation. Random samples are available in App. A. Layout inspired by Karras et al. (2020a).

types of real-world datasets as detailed in Table 1: (1) sets with roto(-reflective) symmetry, such that the image label is invariant under transformation; (2) natural images with preferred orientation (e.g., the `boat` class of images in CIFAR-10 cannot be upside-down). Briefly, they are:

*ANHIR* provides high-resolution pathology slides stained with 5 different dyes to highlight different cellular properties (Borovec et al., 2020; 2018). We extract $128 \times 128$ foreground patches from images of different scales, as described in App. C.1.2. We use the staining dye as conditioning.

*LYSTO* is a multi-organ pathology benchmark for the counting of immunohistochemistry stained lymphocytes (Ciompi et al., 2019). We re-purpose it here for conditional synthesis at a higher resolution of $256 \times 256$. As classification labels are not provided, we use the organ source as class labels. The use of organ sources as classes is validated in App. C.1.1. The high image resolution in addition to the limited sample size of 20,000 make LYSTO a challenging dataset for GANs.

*CIFAR-10* is a natural image vision benchmark of both small resolution and sample size (Krizhevsky et al., 2009). Previous work (Weiler & Cesa, 2019; Romero et al., 2020) finds that equivariant-networks improve classification accuracy on CIFAR-10 and we include here it as a GAN benchmark.

*Food-101* is a small natural image dataset of a 101 categories of food taken in various challenging settings of over/under exposure, label noise, etc. (Bossard et al., 2014). Further, datasets with a high number of classes are known to be challenging for GANs (Odena, 2019). Importantly, even though the objects in this dataset have a preferred pose due to common camera orientations, we speculate that roto-equivariance may be beneficial here as food photography commonly takes an *en face* or mildly oblique view. We resize the training set to $64 \times 64$ resolution for our experiments.

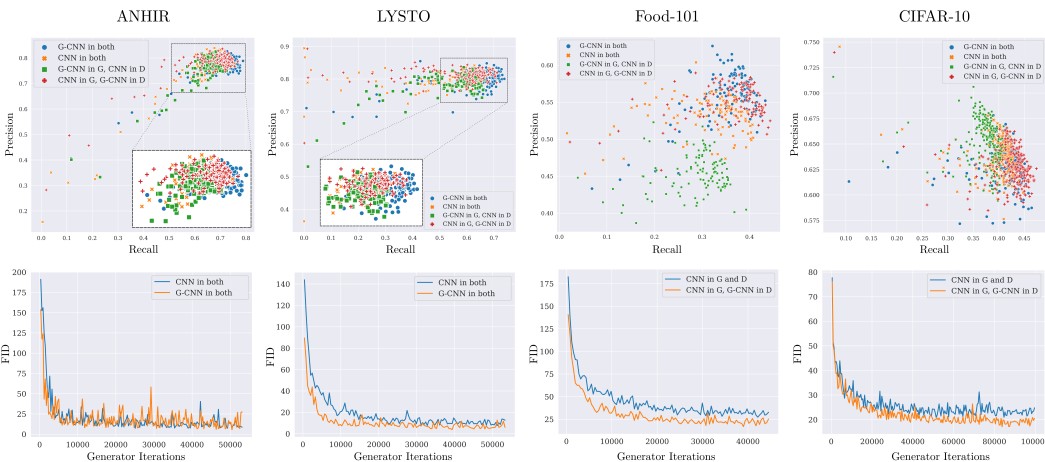

Figure 5: **Top:** Improved Precision and Recall (Kynkäänniemi et al., 2019) analysis of ablations for all snapshots of trained models in each setting (closer to top-right is better). **Bottom:** GAN convergence (FID vs. generator updates) of standard GANs vs. our proposed models for all datasets. For visual clarity, we show only a subset of comparisons with convergence plots for all methods provided in App. A Fig. 7. Readers are encouraged to zoom-in for better inspection.

**Baseline architecture.** To produce a strong non-equivariant baseline, we face several design choices. State-of-the-art GANs follow either BigGAN (Brock et al., 2018) or StyleGAN2 (Karras et al., 2020b) in design. As StyleGAN2 has not yet been demonstrated to scale to *conditional* generation with a large number of classes (to our knowledge), we follow a BigGAN-like construction despite the stability of StyleGAN2. For our small datasets, we make the following modifications: (1) we use fewer channels; (2) we do not use orthogonal regularization; (3) we do not use hierarchical latent projection as we find in early testing that projecting the entire latent to each normalization layer achieves similar results; (4) we do not use attention as equivariant attention is an area of active research (Romero & Hoogendoorn, 2019; Romero et al., 2020) but currently has prohibitively high memory requirements and may not yet scale to GANs. Further details are available in App. C.2.

We then modify either generator (G) and/or discriminator (D) as in Section 2.2 to obtain the corresponding equivariant settings. We note that a discriminator invariant to roto-reflections would assign the same amount of realism to an upright natural image versus a rotated/reflected copy of the same image, allowing the generator to synthesize images at arbitrary orientations. Therefore, for CIFAR-10 and Food-101 we pool over rotations *before* the last residual block to enable the discriminator to detect when generated images are not in their canonical pose while maintaining most of the benefits of equivariance as studied in Weiler & Cesa (2019). We use $p4m$-equivariance for ANHIR and LYSTO and $p4$-equivariance for CIFAR-10 and Food-101 to reduce training time.

**Comparisons.** A natural comparison would be against standard GANs using augmentations drawn from the same group our model is equivariant to. However, augmentation on the real images alone would lead to the augmentations "leaking" into the generated images, e.g., vertical flip augmentation may lead to generated images being upside-down. Zhao et al. (2020c) propose balanced consistency regularization (bCR) for augmentations of both real and generated samples to alleviate this issue, and we thus use it as a comparison. We restrict the augmentations used in bCR to 90-degree rotations or 90-degree rotations and reflections as appropriate to enable a fair comparison against equivariant GANs. Using additional augmentations would help all methods across the board. We further compare against auxiliary rotations (AR) GAN (Chen et al., 2019) where real and fake images are augmented with 90-degree rotations and the discriminator is tasked with predicting their orientation. We do not use AR for ANHIR and LYSTO as they have no canonical orientation. For completeness, we also evaluate standard augmentation (reals only) for all datasets.

**Results.** Quantitative FID results of ablations and comparisons against baselines are presented in Table 3. Equivariant networks (G-CNNs) outperform methods which use standard CNNs with or without augmentation across all datasets. For ANHIR and LYSTO, we find that $p4m$-equivariance in either network improves FID evaluation, with the best results coming from modifying both networks.

However, for the upright datasets CIFAR-10 and Food-101, we find that having a $p4$-equivariant discriminator alone helps more than having both networks be $p4$-equivariant. We speculate that this effect is in part attributable to their orientation bias. With bCR and AR GANs, we find that standard CNNs improve significantly, yet are still outperformed by equivariant nets using no augmentation. We include a mixture of equivariant GANs and bCR for completeness and find that for ANHIR and Food-101, they have an additive effect, whereas they do not for LYSTO and CIFAR-10, indicating a dataset-sensitivity. Of note, we found that bCR with its suggested hyperparameters lead to immediate training collapse on ANHIR, LYSTO, and CIFAR-10, which was fixed by decreasing the strength of the regularization substantially. This may be due to the original work using several different types of augmentation and not just roto-reflections. Standard augmentation (i.e., augmenting training images alone) lead to augmentation leakage for CIFAR-10 and Food-101.

Qualitatively, as class differences in ANHIR should be stain (color) based, we visualize inter-class interpolations between synthesized samples in Figure 3(b). We find that our model better preserves structure while translating between stains, whereas the non-equivariant GAN struggles to do so. In our ablation study in terms of precision and recall in Figure 5, using $p4m$-equivariance in `G` and `D` achieves consistently higher recall for ANHIR and LYSTO. For Food-101, we find that `G-CNN in G and D` achieves higher precision, whereas `CNN in G and G-CNN in D` achieves higher recall. For CIFAR-10 precision and recall, we find no discernable differences between the two settings with lowest FID. Interestingly, for CIFAR-10 adding $p4$-equivariance to `G` but not `D` worsens FID but noticeably improves precision. These observations are consistent with our FID findings as FID tends to correlate better with recall (Karras et al., 2020a). Finally, we plot FID vs. generator updates in Figure 5, finding that the proposed framework converges faster than the baseline as a function of training iterations (for all datasets except ANHIR). Convergence plots for all datasets and all methods compared can be found in App. A Figure 7, showing similar trends.

## 4    DISCUSSION

**Future work.** We present improved conditional image synthesis using equivariant networks, opening several potential future research directions: (1) As efficient implementations of equivariant attention develop, we will incorporate them to model long-range dependency; (2) Equivariance to continuous groups may yield further increased data efficiency and more powerful representations. However, doing so may require non-trivial modifications to current GAN architectures as memory limitations could bottleneck continuous group-equivariant GANs at relevant image sizes. Further, adding more discretizations beyond 4 rotations on a continuous group such has $SE(2)$ may show diminishing returns (Lafarge et al., 2020a, Fig.7); (3) In parallel to our work, Karras et al. (2020a) propose a differentiable augmentation scheme for limited data GANs pertaining to *which* transformations to apply and learning the frequency of augmentation for generic images, with similar work presented in Zhao et al. (2020b). Our approach is fully complementary to these methods when employing transformations outside the considered group and will be integrated into future work; (4) Contemporaneously, Lafarge et al. (2020b) propose equivariant variational autoencoders allowing for control over generated orientations via structured latent spaces which may be used for equivariant GANs as well; (5) The groups considered here do not capture all variabilities present in natural images such as small diffeomorphic warps. Scattering networks may provide an elegant framework to construct GANs equivariant to a wider range of symmetries and enable higher data-efficiency.

**Conclusion.** We present a flexible framework for incorporating symmetry priors within GANs. In doing so, we improve the visual fidelity of GANs in the limited-data regime when trained on symmetric images and even extending to natural images. Our experiments confirm this by improving on conventional GANs across a variety of datasets, ranging from medical imaging modalities to real-world images of food. Modifying either generator or discriminator generally leads to improvements in synthesis, with the latter typically having more impact. To our knowledge, our work is the first to show clear benefits of equivariant learning over standard GAN training on high-resolution conditional image generation beyond toy datasets. While this work is empirical, we believe that it strongly motivates future theoretical analysis of the interplay between GANs and equivariance. Finally, improved results over augmentation-based strategies are presented, demonstrating the benefits of explicit transformation equivariance over equivariance-approximating regularizations.

**Acknowledgements.** Neel Dey thanks Mengwei Ren, Axel Elaldi, Jorge Ono, and Guido Gerig.

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

# A SUPPLEMENTARY RESULTS

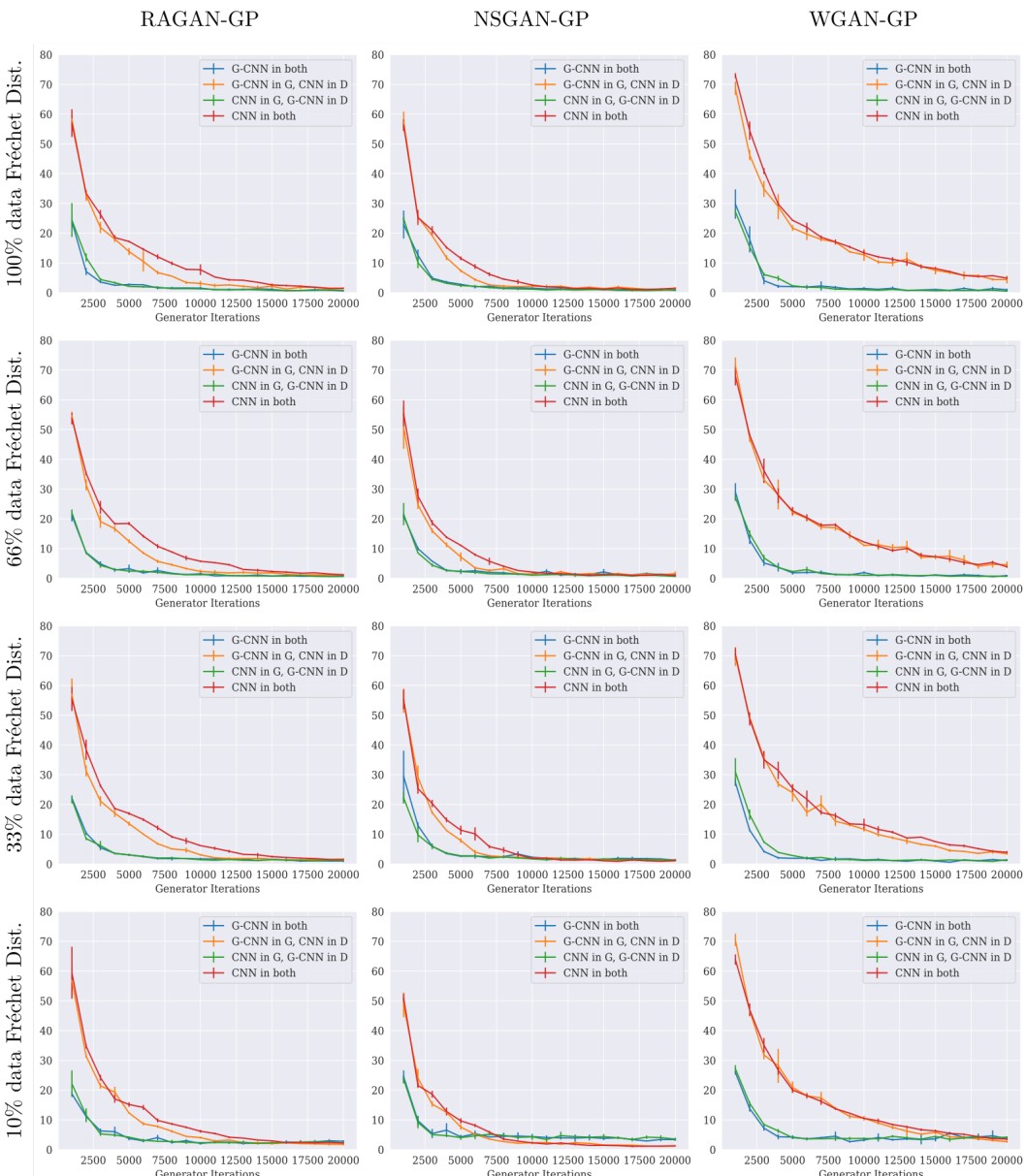

Figure 6: Convergence plots of all GAN ablation settings on Rotated MNIST across data availabilities (rows) and loss functions (columns). Fréchet distance to the validation set is evaluated every 1,000 generator iterations, for 20,000 iterations total. Experiments are repeated with 3 different random seeds and average trajectories are reported with standard deviation error bars. This figure is best interpreted alongside Table 2 which lists best performances for each configuration.

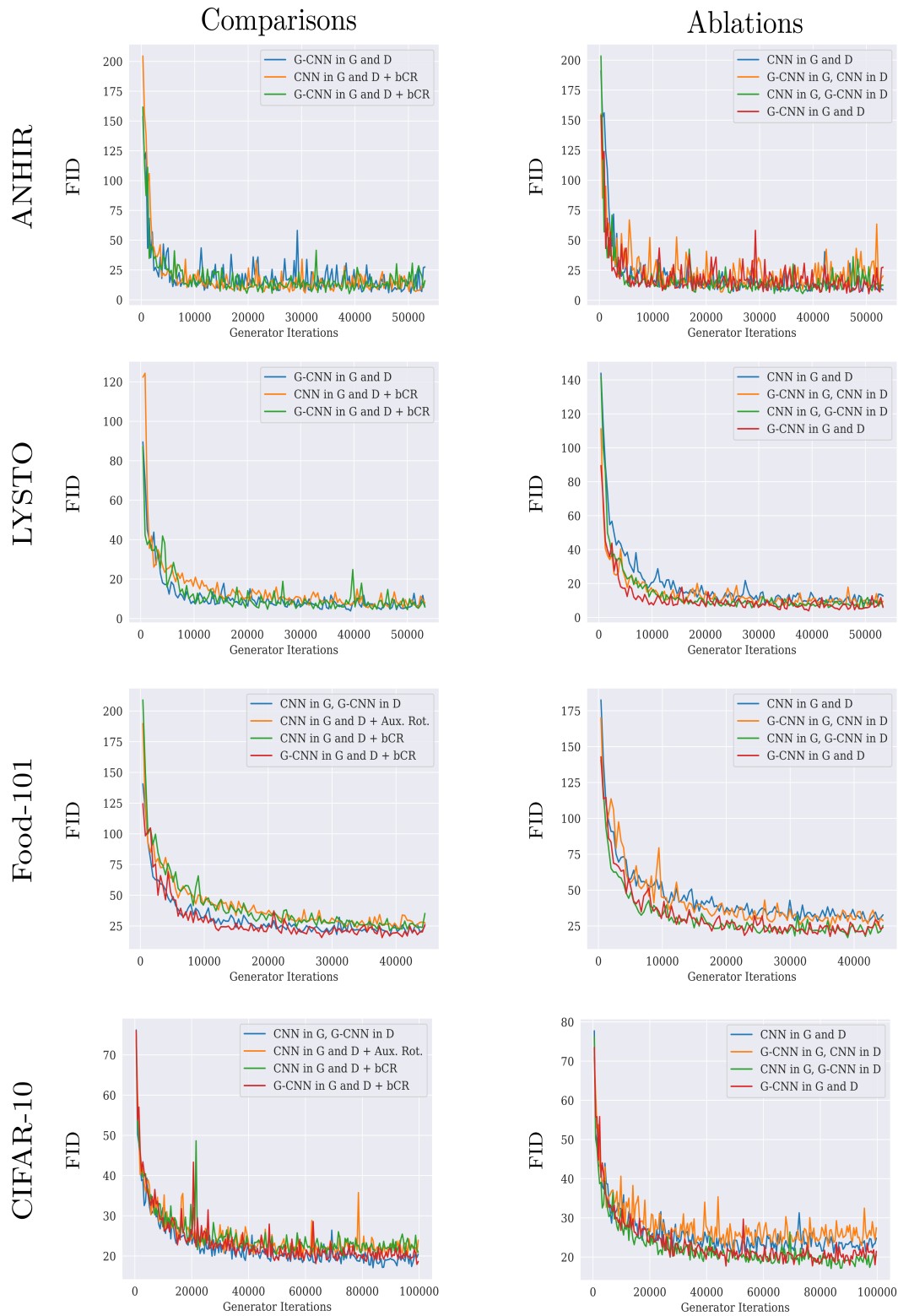

Figure 7: GAN convergence (FID vs. generator updates) for baseline comparisons of the best performing methods (**left**) and ablations (**right**) for all datasets. Readers are encouraged to zoom-in for better inspection.

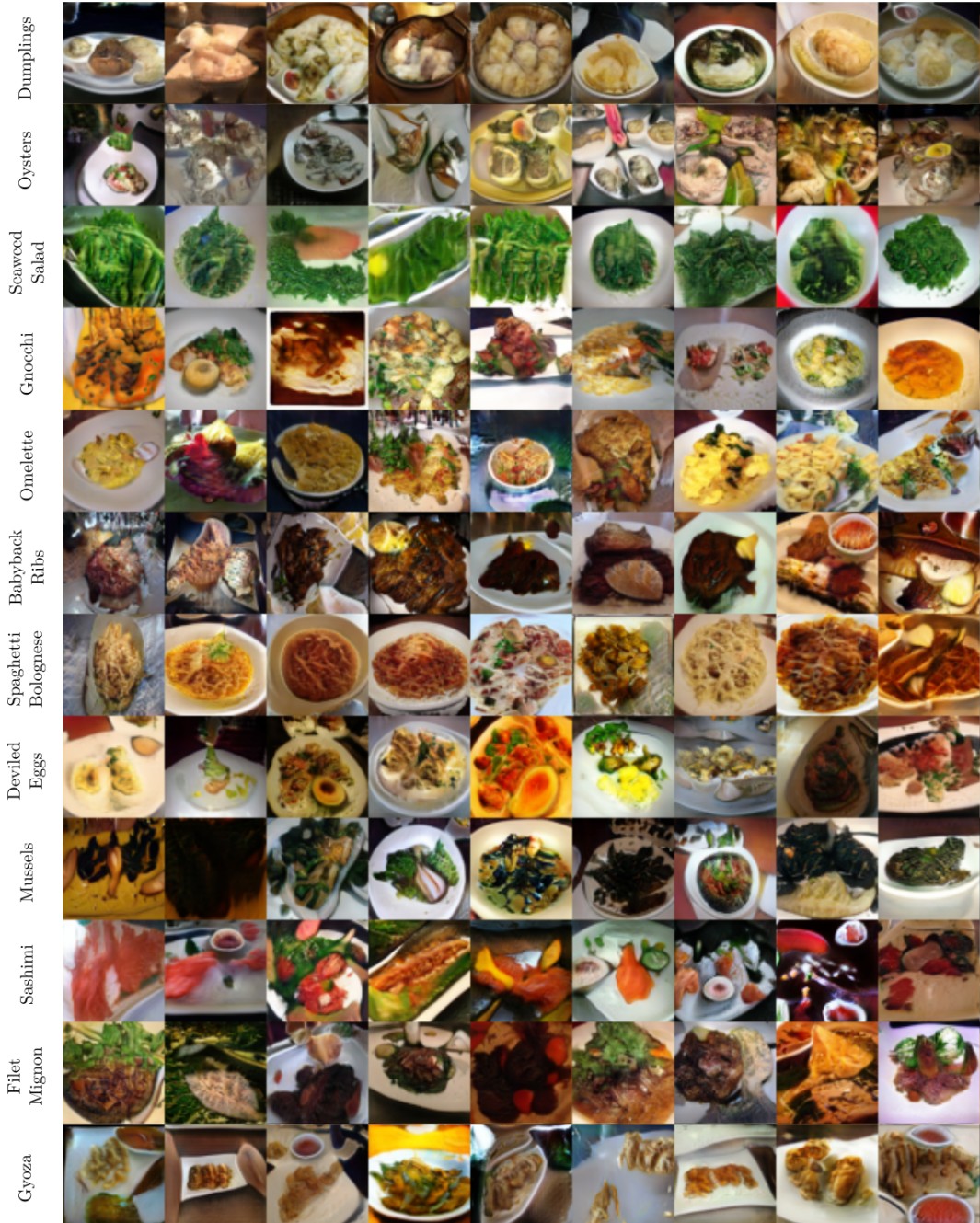

Figure 8: Random $64 \times 64$ Food-101 samples from arbitrarily chosen classes with no truncation taken from the best performing model snapshot with $p4$-equivariance (without augmentation) in the discriminator.

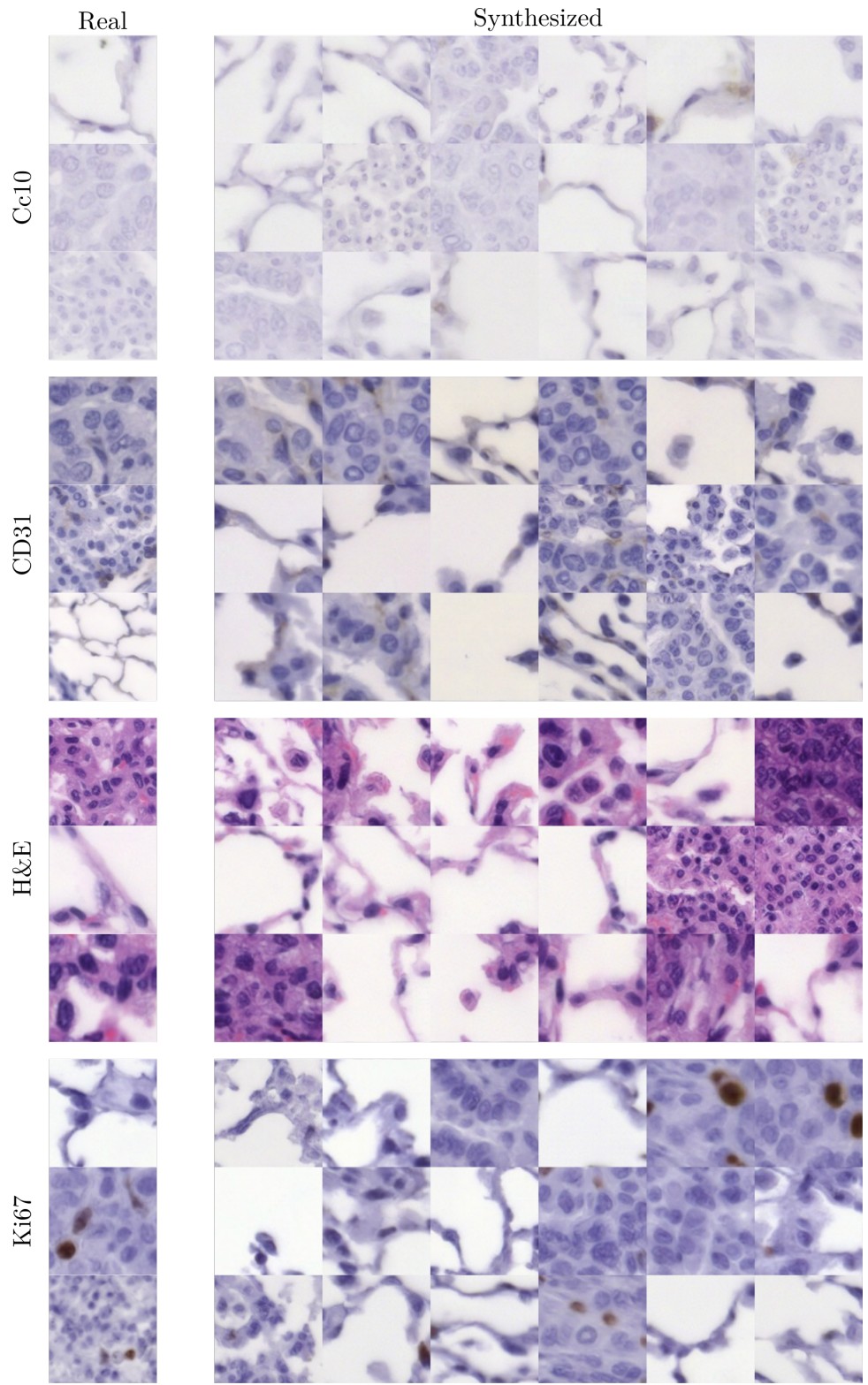

Figure 9: Random $128 \times 128$ ANHIR samples with no truncation taken from the best performing model snapshot with $p4m$-equivariance in both generator and discriminator (without augmentation). Selected real samples are shown in the left column for reference.

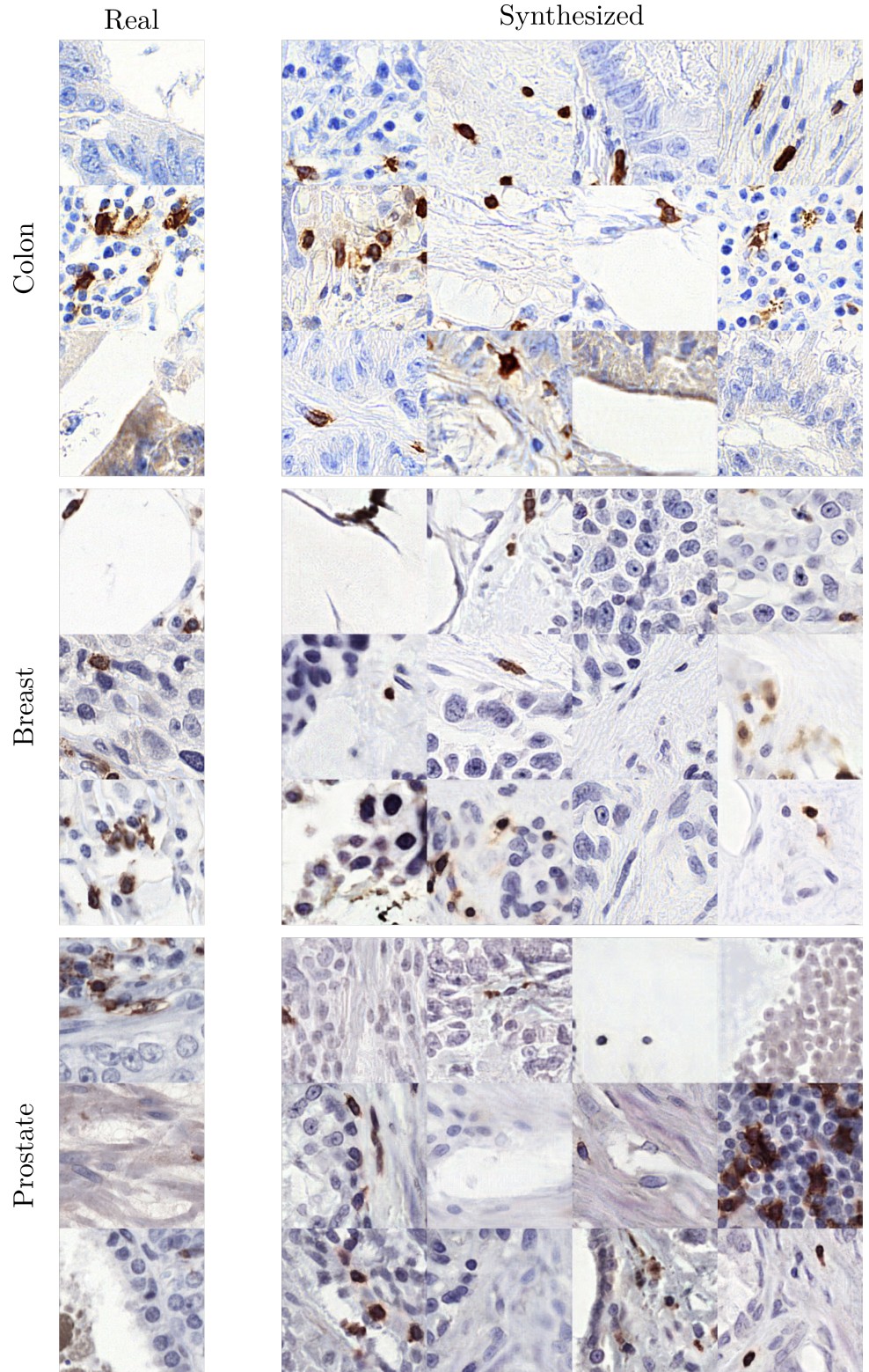

Figure 10: Random $256 \times 256$ LYSTO samples with no truncation taken from the best performing model snapshot with $p4m$-equivariance in both generator and discriminator (without augmentation). Selected real samples are shown in the left column for reference.

(a) RotMNIST                                (b) CIFAR-10

Figure 11: Random samples for RotMNIST ($28 \times 28$) and CIFAR-10 ($32 \times 32$) sampled with $\sigma = 0.75$ truncation trained without augmentation.

Table 4: Kernel Inception Distance results for Map2Sat translation on the Maps dataset. Lower is better.

| Setting | KID |
|---|---|
| Pix2Pix (Isola et al., 2017) | $0.1584 \pm 0.0026$ |
| Pix2Pix (Isola et al., 2017) (optimized) | $0.0663 \pm 0.0038$ |
| CNN in G, G-CNN in D | $0.0333 \pm 0.0005$ |
| G-CNN in G and D | $0.0399 \pm 0.0024$ |

## B  IMAGE-TO-IMAGE TRANSLATION

To show the generic utility of equivariance in generative adversarial network tasks, we present a pilot study employing $p4$-equivariance in supervised image-to-image translation to learn mappings between visual domains. Using the popular `Pix2Pix` model of Isola et al. (2017) as a baseline, we replace both networks with $p4$-equivariant models. For completeness, we also evaluate whether employing $p4$-equivariance in just the discriminator achieves comparable results to modifying both networks, as in the natural image datasets in the main text.

We use the $256 \times 256$ `Maps` dataset first introduced in (Isola et al., 2017), consisting of 1096 training and 1098 validation images of pairs of Google maps images and their corresponding satellite/aerial view images. As FID has a highly biased estimator, its use for evaluating generation with only 1098 validation samples is contraindicated (Binkowski et al., 2018). We instead use the Kernel Inception Distance (KID) proposed by Binkowski et al. (2018) which exhibits low bias for small sample sizes and is adopted in recent image translation studies (Kim et al., 2020). Briefly, as in FID, KID embeds real and fake images into the feature-space of an appropriately chosen network and computes the squared maximum-mean discrepancy (with a polynomial kernel) between their embeddings. Lower values of KID are better. We use the official `Tensorflow` implementation and weights[1].

For baseline `Pix2Pix`, we use pre-trained weights provided by the authors[2]. Interestingly, we find that their architectures can be optimized for improved performance by replacing transposed convolutions with resize-convolutions, reducing the number of parameters by swapping $4 \times 4$ convolutional

---

[1] `https://github.com/tensorflow/gan/blob/master/tensorflow_gan/python/eval/inception_metrics.py`

[2] `https://github.com/junyanz/pytorch-CycleGAN-and-pix2pix`

| Input | Pix2Pix | Ours | Real |
|---|---|---|---|

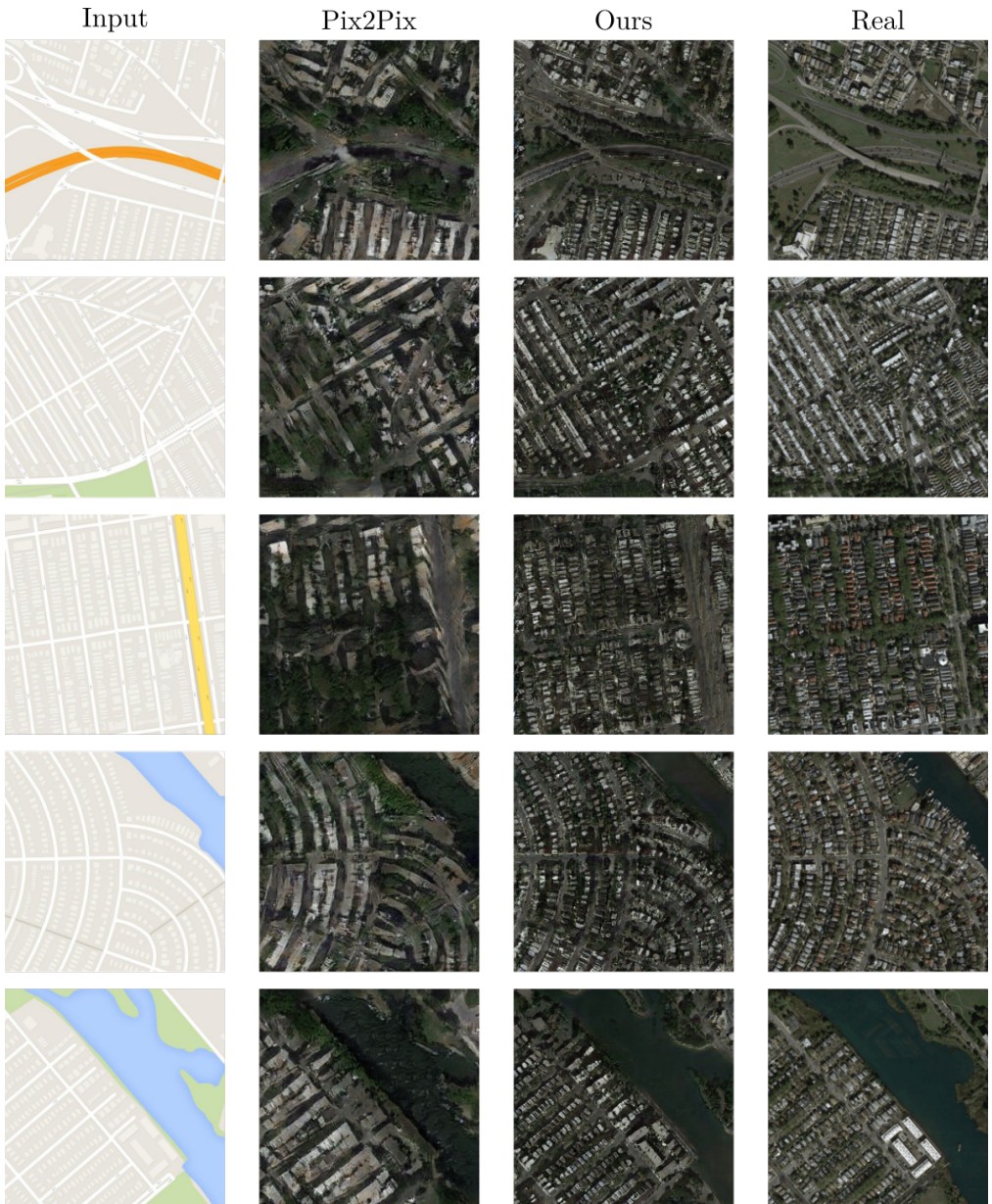

Figure 12: Arbitrarily selected sample translations from input map images (**Col. 1**) using either baseline Pix2Pix with publicly available pre-trained weights (**Col. 2**) or Pix2Pix with a $p4$-equivariant discriminator (**Col. 3**). Real aerial images are shown in Col. 4.

kernels for $3 \times 3$ kernels, and removing dropout. For equivariant models, we replace convolutions with $p4$-convolutions in this optimized architecture and halve the number of filters to keep the number of parameters similar across settings. Architectures are given in Tables 15 and 16. We leave all other experimental details identical to Isola et al. (2017) for all models, such as training for 200 epochs with random crops under a cross-entropy GAN loss.

Quantitative results are presented in Table 4 which shows that $p4$-equivariance in either setting improves over both original baseline and optimized baseline by a wide margin, with the best results coming from $p4$-equivariance in the discriminator alone. Qualitative results are presented in Figure 12 showing improved translation fidelity, further supporting our hypothesis that equivariant networks benefit GAN tasks generically.

## C  Experimental details

### C.1  Data preparation

#### C.1.1  LYSTO class conditioning

To validate the assumption of the organ source being a discriminative feature, a suitable test would be to train a classifier to distinguish between sources. We partition the original training set with a 60/40 train/test split. The original testing set is not used as it has no publicly available organ source information. The dataset has 3 classes - colon, breast, and prostate. Holding out 20% of the new constructed training set for validation, we fine-tune ImageNet-pretrained VGG16 (Simonyan & Zisserman, 2014) and achieve 98% organ classification test accuracy, thus validating our assumption.

#### C.1.2  ANHIR patch extraction

To extract patches for image synthesis, we choose the `lung-lesion` images from the larger AN-HIR dataset, as these images are provided at different scales and possess diverse staining. The images were cropped to the nearest multiples of 128, and $128 \times 128$ patches were then extracted. Foreground/background masking was performed via K-means clustering, followed by morphological dilation. The images were then gridded into $128 \times 128$ patches, i.e., there was no overlap between patches. If a patch contained less than 10% foreground pixels, it was excluded from consideration.

### C.2  Additional Implementation Details

The following subsections list dataset-specific training strategies. Unless noted, all layers use orthogonal initializations. Batch normalization momentum is set to 0.1, and LeakyReLU slopes are set to 0.2 (if used). Spectral normalization is used everywhere except for the dense layer which learns the class embedding as specified in the BigGAN `PyTorch` GitHub repository[3].

For ablation studies, as GANs consist of two networks (the generator and discriminator), we replace group-equivariant layers (convolutional, normalization, and pooling) with the corresponding standard layers in either generator or discriminator to evaluate which network benefits the most from equivariant learning. When we remove equivariant layers from both networks, we recover our baseline comparison. All settings use roughly the same number of parameters, with a very small difference in parameter count arising from the $p4$ (or $p4m$) class-conditional batch normalization layers requiring fewer affine scale and shift parameters than their corresponding standard normalization layers. Tangentially, we note that the equivariant networks require higher amounts of computation time. For example, for a fixed number of training iterations on ANHIR, $p4m$-equivariant GANs currently require approximately four times the amount of computation time.

To identify a common shared stable hyperparameter configuration for all ablations of our method on real datasets, a grid search was performed for the ANHIR dataset over learning rates for generator and discriminator $(\eta_g, \eta_d) : (\{10^{-4}, 4 \times 10^{-4}\}, \{5 \times 10^{-5}, 2 \times 10^{-4}\})$, gradient penalty strengths $(\gamma = \{0.01, 0.1, 1.0, 10.0\})$, and binary choices as to whether to use batch normalization in the discriminator or not, whether to use average-pooling or max-pooling to reduce spatial extent in the discriminator, and whether to use a Gaussian latent space or a Bernoulli latent space. We use the identified hyperparameter configuration as an initial starting point for all datasets, modifying them as appropriate as described below.

For ANHIR, LYSTO, and Food-101 we use the relativistic average adversarial loss (Jolicoeur-Martineau, 2018) for its stability and for CIFAR-10 we use the Hinge loss (Lim & Ye, 2017; Tran et al., 2017) to remain consistent with the literature for that dataset. For our implementation of auxiliary rotations GAN (Chen et al., 2019), we use the suggested regularization weights. For balanced consistency regularization (bCR) (Zhao et al., 2020c), we find that dataset-specific tuning of the regularization strength was required.

---

[3]https://github.com/ajbrock/BigGAN-PyTorch

### C.2.1 ROTMNIST

Given the low resolution of Rotated MNIST, we take a straightforward approach to synthesis without residual connections. In the generator, we sample from a $64D$ Gaussian latent space, concatenate class embeddings, and linearly project as described in Section 2.2. Four spectrally-normalized convolutional layers are then used with class-conditional batch normalization employed after every convolution except for the first and last layer. The discriminator uses three spectrally normalized convolutional layers, with leaky ReLU non-linearities. Average pooling is used to reduce the spatial extent of the feature maps, with global average pooling and conditional projection used at the end of the sequence. For NSGAN and RaGAN, we use the $R_1$ GP, conservatively setting $\gamma = 0.1$. For WGAN, we use the GP defined in Gulrajani et al. (2017) to ensure the 1-Lipschitz constraint with the recommended weight of 10.0. Learning rates were set to $\eta_G = 0.0001$ and $\eta_D = 0.0004$, respectively. For the $p4$-equivariant models, max-pooling over rotations is used after the last group-convolutional layer in both generator and discriminator to get planar feature maps. Architectures are presented in Tables 5 and 6.

### C.2.2 ANHIR

We sample from a $128D$ Gaussian latent space with a batch size of 32. The generator consists of 6 pre-activation residual blocks followed by a final convolutional layer to obtain a 3-channel output. We use class-conditional batch normalization after every convolution, except at the final layer. The discriminator uses 5 pre-activation residual blocks, followed by global average pooling and conditional projection. In the equivariant settings, we use residual blocks with $p4m$-convolutions for roto-reflective symmetries. We train with the relativisitic average loss and use the $R_1$ GP with $\gamma = 0.1$. Learning rates are set to $\eta_G = 0.0001$ and $\eta_D = 0.0004$. All models were trained for approximately 60,000 generator iterations. bCR weights for comparison were set to $\lambda_{real} = 0.1$ and $\lambda_{fake} = 0.05$ for roto-reflective augmentations, with higher values collapsing training. Architectures are presented in Tables 7 and 8.

### C.2.3 LYSTO

Implementation for LYSTO is similar to that of App. C.2.2, with some key differences due to the greater difficulty of training. Due to memory constraints, we use a batch size of 16. We increase the number of residual blocks to 6 in both generator and discriminator and halve the number of filters. The equivariant settings used the $p4m$ roto-reflective symmetries. We initially experienced low sample diversity across a variety of hyperparameter settings. Contrary to recent literature, we find that using batch normalization in the discriminator in addition to spectral normalization greatly improves training for this dataset. Further, halving the learning rates for both networks to $\eta_G = 0.00005$ and $\eta_D = 0.0002$ and increasing the strength of the gradient penalty to 1.0 were necessary for ensuring training stability. As in App. C.2.2, all models were trained for approximately 60,000 generator iterations and bCR weights were set to $\lambda_{real} = 0.1$ and $\lambda_{fake} = 0.05$ for roto-reflective augmentations. As test set labels are not publicly available for LYSTO, we evaluate FID, Precision, and Recall to the training set itself as done in a subset of experiments within Jolicoeur-Martineau (2018) and Zhao et al. (2020b). Architectures are presented in Tables 9 and 10.

### C.2.4 CIFAR-10

For CIFAR-10, we make the following changes to our training parameters to be in accordance with prior art for BigGAN-like designs for this dataset: (1) layer weights are now initialized from $\mathcal{N}(0, 0.02)$; (2) average pooling is used in the discriminator instead of max pooling; (3) learning rates $\eta_G$ and $\eta_D$ are now equal and set to 0.0002; (4) the discriminator is updated four times per generator update; (5) architectures are modified as in Tables 11 and 12; (6) we use the Hinge loss instead of the relativistic average loss. We use a batch size of 64. Karras et al. (2020a) suggest an $R_1$ GP weight of $\gamma = 0.01$ for CIFAR-10 which we use here. We train all CIFAR-10 GANs for 100K generator iterations. bCR weights were set to $\lambda_{real} = 0.1$ and $\lambda_{fake} = 0.1$ for 90-degree rotation augmentations.

For the $p4$-equivariant discriminators, we move the pooling over the group to before the last residual block as stated in the main text. Alternatively, we experimented with using a single additional standard convolutional layer with 32 filters after the $p4$-residual blocks as a lightweight alternative

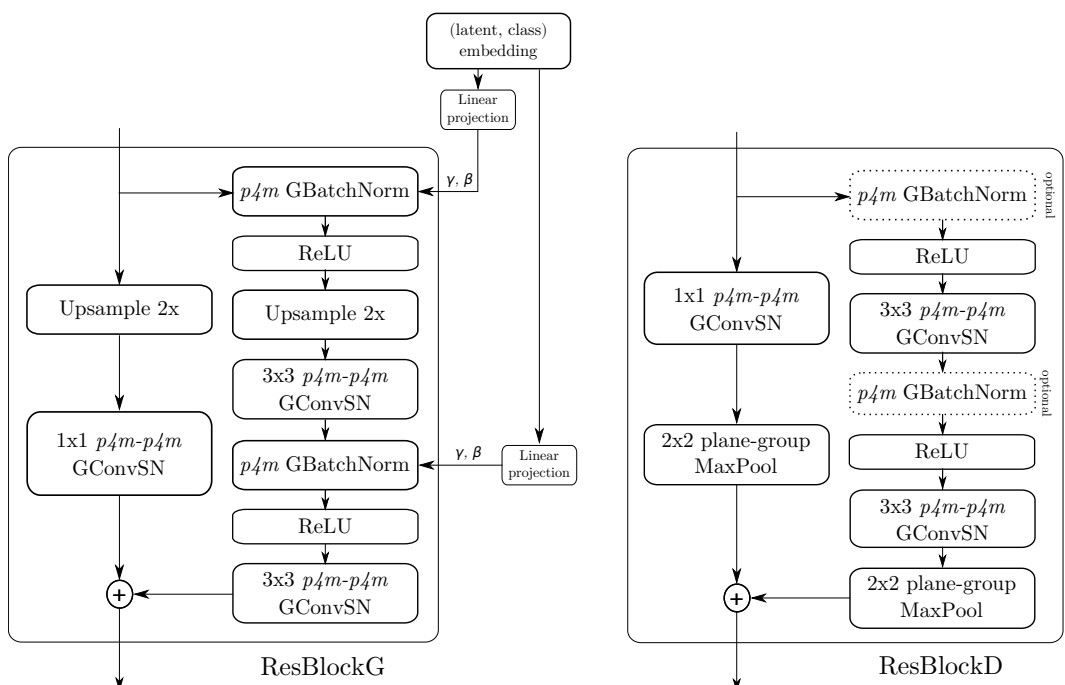

Figure 13: Residual blocks in the group-equivariant settings used in RGB image generation architectures. The choice of $p4$ or $p4m$ is dataset-specific. The generator uses ResBlockG (left) and the discriminator uses ResBlockD (right). The first residual block in the convolutional sequence in either network uses $z2$-$p4m$ group-convolutions for the initial layer. The non-equivariant settings replace all group-convolutions and normalizations within the residual blocks with standard techniques. Visual design inspired by Brock et al. (2018).

to making an entire residual block non-equivariant but this worsened FID evaluation. Interestingly, we find that substituting Global Average Pooling for Global Sum Pooling in the CIFAR-10 discriminators lead to an improvement of $\sim$5 - 8 in terms of FID across the board. This architectural change to the ResNet-based GANs from Gulrajani et al. (2017) was originally made in Miyato et al. (2018), but to our knowledge has not been noted in the literature previously.

### C.2.5 FOOD-101

Compared to the residual synthesis models in App. C.2.2 and C.2.3, we make several changes. We sample from a $64D$ latent Gaussian to lower the number of dense parameters and substantially increase the width of the residual blocks to account for the high number of image classes. We find that an $8\times$ increase in the number of channels for the initial projection from the latent vector and class embedding improves training significantly. We use 4 residual blocks each in both generator and discriminator. For the equivariant setting, we use only $p4$ rotational symmetries to reduce training time. Importantly, we increase the batch size to 64 and the $R_1$ GP to $\gamma = 1.0$, both of which improve the evaluation of all experimental settings. We train all GANs for $\sim$45K generator iterations. The suggested bCR weights of $\lambda_{real} = 10.0$ and $\lambda_{fake} = 10.0$ from Zhao et al. (2020c) were used here for 90-degree rotation augmentations. However, when bCR with default parameters was combined with $p4$-equivariance in G and D, augmentations start to 'leak' into the generated images (e.g., G generating upside-down plates), necessitating lower weights of $\lambda_{real} = 0.5$ and $\lambda_{fake} = 0.5$.

### C.3 ARCHITECTURES

Architectures for the Rotated MNIST experiments are given in Tables 5 and 6, ANHIR in Tables 7 and 8, and LYSTO in Tables 9 and 10. The residual blocks used in the ANHIR, LYSTO, CIFAR-10, and Food-101 experiments are given in Figure 13. SN refers to spectral normalization and $(z2 - p4), (p4 - p4), (z2 - p4m), (p4m - p4m)$ refer to the type of convolution used.

| Generator |
| --- |
| Sample $z \in \mathbb{R}^{64} \sim \mathcal{N}(0, I)$ |
| Embed $y \in \{0, ..., 9\}$ into $\hat{y} \in \mathbb{R}^{64}$ |
| Concatenate $z$ and $\hat{y}$ into $h \in \mathbb{R}^{128}$ |
| Project and reshape $h$ to $7 \times 7 \times 128$ |
| $3 \times 3$ ConvSN, $128 \to 512$ |
| ReLU; Up $2\times$ |
| $3 \times 3$ ConvSN, $512 \to 256$ |
| CCBN$(\cdot, h)$; ReLU; Up $2\times$ |
| $3 \times 3$ ConvSN, $256 \to 128$ |
| CCBN$(\cdot, h)$; ReLU |
| $3 \times 3$ ConvSN, $128 \to 1$ |
| tanh() |

| Discriminator |
| --- |
| Input RGB image $x \in \mathbb{R}^{28 \times 28 \times 1}$ |
| $3 \times 3$ ConvSN, $1 \to 128$ |
| LeakyReLU, Avg. Pool |
| $3 \times 3$ ConvSN, $128 \to 256$ |
| LeakyReLU, Avg. Pool |
| $3 \times 3$ ConvSN, $256 \to 512$ |
| LeakyReLU, Avg. Pool |
| Global Average Pool into $f$ |
| Embed $y \in \{0, ..., 9\}$ into $\hat{y}'$ |
| Projection step$(\hat{y}', f)$ |

Table 5: Architectures used for the standard generator and discriminator in the Rotated MNIST experiments.

| Generator |
| --- |
| Sample $z \in \mathbb{R}^{64} \sim \mathcal{N}(0, I)$ |
| Embed $y \in \{0, ..., 9\}$ into $\hat{y} \in \mathbb{R}^{64}$ |
| Concatenate $z$ and $\hat{y}$ into $h \in \mathbb{R}^{128}$ |
| Project and reshape $h$ to $7 \times 7 \times 128$ |
| $3 \times 3$ $z2 - p4$ GConvSN, $128 \to 256$ |
| ReLU; Up $2\times$ |
| $3 \times 3$ $p4 - p4$ GConvSN, $256 \to 128$ |
| CCBN$(\cdot, h)$; ReLU; Up $2\times$ |
| $3 \times 3$ $p4 - p4$ GConvSN, $128 \to 64$ |
| CCBN$(\cdot, h)$; ReLU |
| $3 \times 3$ $p4 - p4$ GConvSN, $64 \to 1$ |
| $p4$-Max Pool |
| tanh() |

| Discriminator |
| --- |
| Input RGB image $x \in \mathbb{R}^{28 \times 28 \times 1}$ |
| $3 \times 3$ $z2 - p4$ GConvSN, $1 \to 64$ |
| LeakyReLU, Avg. Pool |
| $3 \times 3$ $p4 - p4$ GConvSN, $64 \to 128$ |
| LeakyReLU, Avg. Pool |
| $3 \times 3$ $p4 - p4$ GConvSN, $128 \to 256$ |
| LeakyReLU, Avg. Pool |
| $p4$-Max Pool |
| Global Average Pool into $f$ |
| Embed $y \in \{0, ..., 9\}$ into $\hat{y}'$ |
| Projection step$(\hat{y}', f)$ |

Table 6: Architectures used for the $p4$-equivariant generator and discriminator in the Rotated MNIST experiments.

| Generator |
| --- |
| Sample $z \in \mathbb{R}^{128} \sim \mathcal{N}(0, I)$ |
| Embed $y \in \{0, ..., 4\}$ into $\hat{y} \in \mathbb{R}^{128}$ |
| Concatenate $z$ and $\hat{y}$ into $h \in \mathbb{R}^{256}$ |
| Project and reshape $h$ to $4 \times 4 \times 128$ |
| $z2 - z2$ ResBlockG, $128 \to 512$ |
| $z2 - z2$ ResBlockG, $512 \to 256$ |
| $z2 - z2$ ResBlockG, $256 \to 128$ |
| $z2 - z2$ ResBlockG, $128 \to 64$ |
| $z2 - z2$ ResBlockG, $64 \to 32$ |
| BN; ReLU |
| $3 \times 3$ ConvSN, $32 \to 3$ |
| tanh() |

| Discriminator |
| --- |
| Input RGB image $x \in \mathbb{R}^{128 \times 128 \times 3}$ |
| $z2 - z2$ ResBlockD, $3 \to 32$ |
| $z2 - z2$ ResBlockD, $32 \to 64$ |
| $z2 - z2$ ResBlockD, $64 \to 128$ |
| $z2 - z2$ ResBlockD, $128 \to 256$ |
| $z2 - z2$ ResBlockD, $256 \to 512$ |
| ReLU |
| Global Average Pool into $f$ |
| Embed $y \in \{0, ..., 4\}$ into $\hat{y}'$ |
| Projection step$(\hat{y}', f)$ |

Table 7: Architectures used for the standard generator and discriminator in the ANHIR experiments.

| Generator |
| --- |
| Sample $z \in \mathbb{R}^{128} \sim \mathcal{N}(0, I)$ |
| Embed $y \in \{0, ..., 4\}$ into $\hat{y} \in \mathbb{R}^{128}$ |
| Concatenate $z$ and $\hat{y}$ into $h \in \mathbb{R}^{256}$ |
| Project and reshape $h$ to $4 \times 4 \times 128$ |
| $z2 - p4m$ ResBlockG, $128 \rightarrow 181$ |
| $p4m - p4m$ ResBlockG, $181 \rightarrow 90$ |
| $p4m - p4m$ ResBlockG, $90 \rightarrow 45$ |
| $p4m - p4m$ ResBlockG, $45 \rightarrow 22$ |
| $p4m - p4m$ ResBlockG, $22 \rightarrow 11$ |
| $p4m$-BN; ReLU |
| $3 \times 3$ $p4m - p4m$ GConvSN, $11 \rightarrow 3$ |
| $p4m$-Max Pool |
| tanh() |

| Discriminator |
| --- |
| Input RGB image $x \in \mathbb{R}^{128 \times 128 \times 3}$ |
| $z2 - p4m$ ResBlockD, $3 \rightarrow 11$ |
| $p4m - p4m$ ResBlockD, $11 \rightarrow 22$ |
| $p4m - p4m$ ResBlockD, $22 \rightarrow 45$ |
| $p4m - p4m$ ResBlockD, $45 \rightarrow 90$ |
| $p4m - p4m$ ResBlockD, $90 \rightarrow 181$ |
| ReLU |
| $p4m$-Max Pool |
| Global Average Pool into $f$ |
| Embed $y \in \{0, ..., 4\}$ into $\hat{y}'$ |
| Projection step$(\hat{y}', f)$ |

Table 8: Architectures used for the $p4m$-equivariant generator and discriminator in the ANHIR experiments.

| Generator |
| --- |
| Sample $z \in \mathbb{R}^{128} \sim \mathcal{N}(0, I)$ |
| Embed $y \in \{0, 1, 2\}$ into $\hat{y} \in \mathbb{R}^{128}$ |
| Concatenate $z$ and $\hat{y}$ into $h \in \mathbb{R}^{256}$ |
| Project and reshape $h$ to $4 \times 4 \times 128$ |
| $z2 - z2$ ResBlockG, $128 \rightarrow 512$ |
| $z2 - z2$ ResBlockG, $512 \rightarrow 256$ |
| $z2 - z2$ ResBlockG, $256 \rightarrow 128$ |
| $z2 - z2$ ResBlockG, $128 \rightarrow 64$ |
| $z2 - z2$ ResBlockG, $64 \rightarrow 32$ |
| $z2 - z2$ ResBlockG, $32 \rightarrow 16$ |
| BN; ReLU |
| $3 \times 3$ ConvSN, $16 \rightarrow 3$ |
| tanh() |

| Discriminator |
| --- |
| Input RGB image $x \in \mathbb{R}^{256 \times 256 \times 3}$ |
| $z2 - z2$ ResBlockD-BN, $3 \rightarrow 16$ |
| $z2 - z2$ ResBlockD-BN, $16 \rightarrow 32$ |
| $z2 - z2$ ResBlockD-BN, $32 \rightarrow 64$ |
| $z2 - z2$ ResBlockD-BN, $64 \rightarrow 128$ |
| $z2 - z2$ ResBlockD-BN, $128 \rightarrow 256$ |
| $z2 - z2$ ResBlockD-BN, $256 \rightarrow 512$ |
| ReLU |
| Global Average Pool into $f$ |
| Embed $y \in \{0, 1, 2\}$ into $\hat{y}'$ |
| Projection step$(\hat{y}', f)$ |

Table 9: Architectures used for the standard generator and discriminator in the LYSTO experiments.

| Generator |
| --- |
| Sample $z \in \mathbb{R}^{128} \sim \mathcal{N}(0, I)$ |
| Embed $y \in \{0, 1, 2\}$ into $\hat{y} \in \mathbb{R}^{128}$ |
| Concatenate $z$ and $\hat{y}$ into $h \in \mathbb{R}^{256}$ |
| Project and reshape $h$ to $4 \times 4 \times 128$ |
| $z2 - p4m$ ResBlockG, $128 \rightarrow 181$ |
| $p4m - p4m$ ResBlockG, $181 \rightarrow 90$ |
| $p4m - p4m$ ResBlockG, $90 \rightarrow 45$ |
| $p4m - p4m$ ResBlockG, $45 \rightarrow 22$ |
| $p4m - p4m$ ResBlockG, $22 \rightarrow 11$ |
| $p4m - p4m$ ResBlockG, $11 \rightarrow 5$ |
| $p4m$-BN; ReLU |
| $3 \times 3$ $p4m - p4m$ GConvSN, $5 \rightarrow 3$ |
| $p4m$-Max Pool |
| tanh() |

| Discriminator |
| --- |
| Input RGB image $x \in \mathbb{R}^{256 \times 256 \times 3}$ |
| $z2 - p4m$ ResBlockD-BN, $3 \rightarrow 5$ |
| $p4m - p4m$ ResBlockD-BN, $5 \rightarrow 11$ |
| $p4m - p4m$ ResBlockD-BN, $11 \rightarrow 22$ |
| $p4m - p4m$ ResBlockD-BN, $22 \rightarrow 45$ |
| $p4m - p4m$ ResBlockD-BN, $45 \rightarrow 90$ |
| $p4m - p4m$ ResBlockD-BN, $90 \rightarrow 181$ |
| ReLU |
| $p4m$-Max Pool |
| Global Average Pool into $f$ |
| Embed $y \in \{0, 1, 2\}$ into $\hat{y}'$ |
| Projection step$(\hat{y}', f)$ |

Table 10: Architectures used for the $p4m$-equivariant generator and discriminator in the LYSTO experiments.

| Generator |
|---|
| Sample $z \in \mathbb{R}^{128} \sim \mathcal{N}(0, I)$ |
| Embed $y \in \{0, ..., 9\}$ into $\hat{y} \in \mathbb{R}^{128}$ |
| Concatenate $z$ and $\hat{y}$ into $h \in \mathbb{R}^{256}$ |
| Project and reshape $h$ to $4 \times 4 \times 256$ |
| $z2 - z2$ ResBlockG, $256 \rightarrow 256$ |
| $z2 - z2$ ResBlockG, $256 \rightarrow 256$ |
| $z2 - z2$ ResBlockG, $256 \rightarrow 256$ |
| BN; ReLU |
| $3 \times 3$ ConvSN, $256 \rightarrow 3$ |
| tanh() |

| Discriminator |
|---|
| Input RGB image $x \in \mathbb{R}^{32 \times 32 \times 3}$ |
| $z2 - z2$ ResBlockD (avg. pool), $3 \rightarrow 128$ |
| $z2 - z2$ ResBlockD (avg. pool), $128 \rightarrow 128$ |
| $z2 - z2$ ResBlockD (no downsample), $128 \rightarrow 128$ |
| $z2 - z2$ ResBlockD (no downsample), $128 \rightarrow 128$ |
| ReLU |
| Global Sum Pool into $f$ |
| Embed $y \in \{0, ..., 9\}$ into $\hat{y}'$ |
| Projection step$(\hat{y}', f)$ |

Table 11: Architectures used for the standard generator and discriminator in the CIFAR-10 experiments.

| Generator |
|---|
| Sample $z \in \mathbb{R}^{128} \sim \mathcal{N}(0, I)$ |
| Embed $y \in \{0, ..., 9\}$ into $\hat{y} \in \mathbb{R}^{128}$ |
| Concatenate $z$ and $\hat{y}$ into $h \in \mathbb{R}^{256}$ |
| Project and reshape $h$ to $4 \times 4 \times 256$ |
| $z2 - p4$ ResBlockG, $256 \rightarrow 128$ |
| $p4 - p4$ ResBlockG, $128 \rightarrow 128$ |
| $p4 - p4$ ResBlockG, $128 \rightarrow 128$ |
| $p4$-BN; ReLU |
| $3 \times 3 \, p4 - p4$ GConvSN, $128 \rightarrow 3$ |
| $p4$-Max Pool |
| tanh() |

| Discriminator |
|---|
| Input RGB image $x \in \mathbb{R}^{64 \times 64 \times 3}$ |
| $z2 - p4$ ResBlockD (avg. pool), $3 \rightarrow 64$ |
| $z2 - p4$ ResBlockD (avg. pool), $64 \rightarrow 64$ |
| $p4 - p4$ ResBlockD (no downsample), $64 \rightarrow 64$ |
| $p4$-Max Pool |
| $z2 - z2$ ResBlockD (no downsample), $64 \rightarrow 128$ |
| ReLU |
| Global Sum Pool into $f$ |
| Embed $y \in \{0, ..., 9\}$ into $\hat{y}'$ |
| Projection step$(\hat{y}', f)$ |

Table 12: Architectures used for the $p4$-equivariant generator and discriminator in the CIFAR-10 experiments.

| Generator |
|---|
| Sample $z \in \mathbb{R}^{64} \sim \mathcal{N}(0, I)$ |
| Embed $y \in \{0, ..., 100\}$ into $\hat{y} \in \mathbb{R}^{64}$ |
| Concatenate $z$ and $\hat{y}$ into $h \in \mathbb{R}^{128}$ |
| Project and reshape $h$ to $4 \times 4 \times 1024$ |
| $z2 - z2$ ResBlockG, $1024 \rightarrow 512$ |
| $z2 - z2$ ResBlockG, $512 \rightarrow 384$ |
| $z2 - z2$ ResBlockG, $384 \rightarrow 256$ |
| $z2 - z2$ ResBlockG, $256 \rightarrow 192$ |
| BN; ReLU |
| $3 \times 3$ ConvSN, $192 \rightarrow 3$ |
| tanh() |

| Discriminator |
|---|
| Input RGB image $x \in \mathbb{R}^{64 \times 64 \times 3}$ |
| $z2 - z2$ ResBlockD, $3 \rightarrow 128$ |
| $z2 - z2$ ResBlockD, $128 \rightarrow 256$ |
| $z2 - z2$ ResBlockD, $256 \rightarrow 512$ |
| $z2 - z2$ ResBlockD, $512 \rightarrow 784$ |
| ReLU |
| Global Average Pool into $f$ |
| Embed $y \in \{0, ..., 100\}$ into $\hat{y}'$ |
| Projection step$(\hat{y}', f)$ |

Table 13: Architectures used for the standard generator and discriminator in the Food-101 experiments.

| Generator |
| --- |
| Sample $z \in \mathbb{R}^{64} \sim \mathcal{N}(0, I)$ |
| Embed $y \in \{0, ..., 100\}$ into $\hat{y} \in \mathbb{R}^{64}$ |
| Concatenate $z$ and $\hat{y}$ into $h \in \mathbb{R}^{128}$ |
| Project and reshape $h$ to $4 \times 4 \times 1024$ |
| $z2 - p4$ ResBlockG, $1024 \to 256$ |
| $p4 - p4$ ResBlockG, $256 \to 192$ |
| $p4 - p4$ ResBlockG, $192 \to 128$ |
| $p4 - p4$ ResBlockG, $128 \to 96$ |
| $p4$-BN; ReLU |
| $3 \times 3\ p4 - p4$ GConvSN, $96 \to 3$ |
| $p4$-Max Pool |
| tanh() |

| Discriminator |
| --- |
| Input RGB image $x \in \mathbb{R}^{64 \times 64 \times 3}$ |
| $z2 - p4$ ResBlockD, $3 \to 64$ |
| $p4 - p4$ ResBlockD, $64 \to 128$ |
| $p4 - p4$ ResBlockD, $128 \to 256$ |
| $p4$-Max Pool |
| $z2 - z2$ ResBlockD, $256 \to 784$ |
| ReLU |
| Global Average Pool into $f$ |
| Embed $y \in \{0, ..., 100\}$ into $\hat{y}'$ |
| Projection step($\hat{y}', f$) |

Table 14: Architectures used for the $p4$-equivariant generator and discriminator in the Food-101 experiments.

| Generator |
| --- |
| Input RGB image $x \in \mathbb{R}^{256 \times 256 \times 3}$ |
| $h_1 : z2 - z2$ DownBlock, $3 \to 64$ |
| $h_2 : z2 - z2$ DownBlock, $64 \to 128$ |
| $h_3 : z2 - z2$ DownBlock, $128 \to 256$ |
| $h_4 : z2 - z2$ DownBlock, $512 \to 512$ |
| $h_5 : z2 - z2$ DownBlock, $512 \to 512$ |
| $h_6 : z2 - z2$ DownBlock, $512 \to 512$ |
| $h_7 : z2 - z2$ DownBlock, $512 \to 512$ |
| $h_8 : z2 - z2$ DownBlock, $512 \to 512$ |
| $z2 - z2$ UpBlock, $512 \to 512$; Concatenate $h_7$ |
| $z2 - z2$ UpBlock, $512 \to 512$; Concatenate $h_6$ |
| $z2 - z2$ UpBlock, $512 \to 512$; Concatenate $h_5$ |
| $z2 - z2$ UpBlock, $512 \to 512$; Concatenate $h_4$ |
| $z2 - z2$ UpBlock, $512 \to 256$; Concatenate $h_3$ |
| $z2 - z2$ UpBlock, $256 \to 128$; Concatenate $h_2$ |
| $z2 - z2$ UpBlock, $128 \to 64$; Concatenate $h_1$ |
| Upsample $2\times$, $3 \times 3$ Conv, $64 \to 3$ |
| tanh() |

| Discriminator |
| --- |
| Input RGB image $x \in \mathbb{R}^{256 \times 256 \times 3}$ |
| Input RGB image $y \in \mathbb{R}^{256 \times 256 \times 3}$ |
| Concatenate $x$ and $y$ feature-wise |
| $z2 - z2$ DownBlock, $3 \to 64$ |
| $z2 - z2$ DownBlock, $64 \to 128$ |
| $z2 - z2$ DownBlock, $128 \to 256$ |
| $z2 - z2\ 3 \times 3$ Conv $256 \to 512$, BatchNorm, Leaky ReLU |
| $z2 - z2\ 3 \times 3$ Conv $512 \to 1$, tanh() |

Table 15: Architectures used for the standard generator and discriminator in the Pix2Pix experiments. Each DownBlock consists of a $3 \times 3$ Convolution, $2\times$ Average Pool, Batch Normalization, and Leaky ReLU activation. Each UpBlock consists of $2\times$ nearest-neighbors upsampling, $3 \times 3$ Convolution, Batch Normalization, and Leaky ReLU activation.

| Generator |
|---|
| Input RGB image $x \in \mathbb{R}^{256 \times 256 \times 3}$ |
| $h_1 : z2 - p4$ DownBlock, $3 \to 64$ |
| $h_2 : p4 - p4$ DownBlock, $64 \to 128$ |
| $h_3 : p4 - p4$ DownBlock, $128 \to 256$ |
| $h_4 : p4 - p4$ DownBlock, $512 \to 512$ |
| $h_5 : p4 - p4$ DownBlock, $512 \to 512$ |
| $h_6 : p4 - p4$ DownBlock, $512 \to 512$ |
| $h_7 : p4 - p4$ DownBlock, $512 \to 512$ |
| $h_8 : p4 - p4$ DownBlock, $512 \to 512$ |
| $p4 - p4$ UpBlock, $512 \to 512$; Concatenate $h_7$ |
| $p4 - p4$ UpBlock, $512 \to 512$; Concatenate $h_6$ |
| $p4 - p4$ UpBlock, $512 \to 512$; Concatenate $h_5$ |
| $p4 - p4$ UpBlock, $512 \to 512$; Concatenate $h_4$ |
| $p4 - p4$ UpBlock, $512 \to 256$; Concatenate $h_3$ |
| $p4 - p4$ UpBlock, $256 \to 128$; Concatenate $h_2$ |
| $p4 - p4$ UpBlock, $128 \to 64$; Concatenate $h_1$ |
| Upsample $2\times$, $3 \times 3$ GConv, $64 \to 3$ |
| $p4$-average pool, tanh() |

| Discriminator |
|---|
| Input RGB image $x \in \mathbb{R}^{256 \times 256 \times 3}$ Input RGB image $y \in \mathbb{R}^{256 \times 256 \times 3}$ Concatenate $x$ and $y$ feature-wise |
| $z2 - p4$ DownBlock, $3 \to 64$ |
| $p4 - p4$ DownBlock, $64 \to 128$ |
| $p4 - p4$ DownBlock, $128 \to 256$ |
| $p4 - p4$ $3 \times 3$ Conv $256 \to 512$, BatchNorm, Leaky ReLU |
| $p4 - p4$ $3 \times 3$ Conv $512 \to 1$, $p4$-average pool |
| tanh() |

Table 16: Architectures used for the $p4$-equivariant generator and discriminator in the Pix2Pix experiments. Each DownBlock consists of a $3 \times 3$ $p4$-convolution, $2\times$ Average Pool, $p4$-Batch Normalization, and Leaky ReLU activation. Each UpBlock consists of $2\times$ nearest-neighbors upsampling, $3 \times 3$ $p4$-Convolution, $p4$-Batch Normalization, and Leaky ReLU activation.

