# OpenReview forum: "Group Equivariant Generative Adversarial Networks"
_ICLR.cc/2021/Conference — ICLR 2021 Poster_

### Official Review · AnonReviewer3 · 2020-10-27
**Interesting application of G-CNNs to image synthesis, but limited novelty.**

**Rating:** 6
**Confidence:** 4

**Review:**

Summary:
The submission concerns an application of group convolutions (Cohen & Welling, 2016) to the image synthesis setting, where images are produced by the generator of a GAN. The two GAN components are augmented mainly by a straightforward replacement of "regular" convolutions by group convolutions, in addition to some other training tricks of the trade (gradient penalty, spectral normalization). Experiments indicate somewhat lower FID scores on both synthetic and real settings. The method is seen as useful especially for the low data regime case.

Review:
Despite the conceptual simplicity of the presented approach ("replace convolutions by group convolutions") I found the method itself a novel combination of existing concepts. I can also imagine that getting the such modified architecture(s) to work in practice can be quite tricky; hence the additional GAN training measures that were adopted in the training setup.

Overall, however, I see novelty and impact to be fairly limited, as all insights come from empirical evaluation, which is notoriously difficult for visual synthesis applications. The authors provide both visual results as well as quantitative results mostly on the basis of FID measurements which aim to evaluate the quality of the images. Experiments seem sound and results are clearly presented.

Although the authors claim in the discussion section that visual fidelity and sample complexity are meaningfully improved, I miss an attempt at quantitative analysis of this claim, either through algorithmic metrics or user studies. There are also no statements on the trade-off between sample fidelity, training difficult and amount of (training or inference time) compute. Since one of the strengths of the approach may be in the limited data regime, I would have liked to see stronger evidence of a major impact there; I can't quite see such a trend in the numbers of Table 1.

Overall I see the work performed and results achieved by the submission as good, but a stronger verification of the main claims would make the submission even stronger for the main conference track.

[Update post-rebuttal: I thank the authors for addressing some of the concerns raised by the reviewers. My stance remains, also given the outcome of e.g. the 10% experiment -- verification of the main claims remains difficult. My score already reflects that I'd be happy to see the submission accepted.]

---

> ### Author Response · Authors · 2020-11-23
> **Reviewer 3 Response (part 1)**
>
> We appreciate and thank you for your thoughtful assessment and useful feedback. We hope to address your comments below:
>
> > _Despite the conceptual simplicity of the presented approach, I found the method itself a novel combination of existing concepts. I can also imagine that getting the such modified architecture(s) to work in practice can be quite tricky_
>
> Thank you for highlighting the novelty and practical difficulty of the work.
>
> > _Overall, however, I see novelty and impact to be fairly limited, as all insights come from empirical evaluation, which is notoriously difficult for visual synthesis applications. The authors provide both visual results as well as quantitative results mostly on the basis of FID measurements which aim to evaluate the quality of the images. Experiments seem sound and results are clearly presented._
>
> We fully agree that visual synthesis evaluation is quite tricky in practice. A user study-based evaluation would be ideal but is typically not performed for GAN studies. The literature has converged around evaluating generative visual models via algorithmic distances and we follow this trend. We now acknowledge the limitations of this approach in Section 3/Experimental Methodologies.
>
> > _Although the authors claim in the discussion section that visual fidelity and sample complexity are meaningfully improved, I miss an attempt at quantitative analysis of this claim, either through algorithmic metrics or user studies._
>
> [As reviewer 1 raised a similar point, we post the same response here] This is a miscommunication on our part. To clarify, we claim that the real world datasets we use are themselves in the limited data regime. Class-conditional GANs are typically trained on consisting of hundreds of thousands to millions of samples on datasets such as ImageNet (1M samples) and LSUN (several million samples), whereas ANHIR, LYSTO, Food-101, and CIFAR-10 have 28407, 20000, 75747, and 50000 training images, respectively. Our largest dataset of 75K samples (Food-101) has 101 categories and GAN training difficulty is known to increase with higher numbers of classes [Odena19], and CIFAR-10 has recently been shown to be a limited data benchmark for GANs as well in contemporaneous work [Karras20]. As a highlight, we improve the baseline FID by 10+ points for Food-101 in the limited regime using roto-translation equivariant networks. We accordingly rephrase our sample complexity claims in the paper for clarity.
>
> > _Since one of the strengths of the approach may be in the limited data regime, I would have liked to see stronger evidence of a major impact there; I can't quite see such a trend in the numbers of Table 1._
>
> Regarding the referenced table (Table 2 in the revision): we use Rotated MNIST (12k training, 50k testing) as a toy dataset for GAN evaluation as it is universally used in equivariant learning papers for benchmarking purposes. Our experiments indicate that Rotated MNIST may be an "easy" dataset for the stabilized GAN setup we employ as FID results are relatively insensitive to data availability. We do notice that minimum FID tends to go down as more data becomes available. Other trends we notice are more apparent from Figure 6, where using G-CNNs over CNNs in the discriminator (or in both generator and discriminator) significantly speeds up convergence and generally achieves lower FID overall as shown in Table 1, both of which are consistent with our driving hypothesis.
>
> We note that FID results across data availability should be interpreted with some caution, as FID has a biased estimator [Bińkowski18]. To ameliorate these concerns, we always generate the same number of samples as the validation set (50K) regardless of whether the training set had all samples (12K) or only 10% (1200). In row 1 of Table 2, we now add ideal FID scores for that percentage of data availability by computing the FID between the training set (sampled with replacement) to the validation set averaged over 100 runs, showing that as data availability decreases, the ideal FID worsens.
>
> In addition, as suggested by Rev. 1, we now include a 10% data experiment where we see that using a G-CNN in the generator may be of more utility than using a G-CNN in the discriminator. We hypothesize that this is due to using the same hyperparameters in all data quantity settings, leading to suboptimal training scenarios such as a batch size of 64 with 1200 samples, leading to the discriminator overpowering the generator at 10%. In the revision, we repeat all Rotated MNIST experiments with 3 random seeds and report average statistics. We now expand our discussion of the Rotated MNIST dataset results in Section 3.1.

---

> > ### Author Response · Authors · 2020-11-23
> > **Reviewer 3 Response (part 2)**
> >
> > (continued from above)
> >
> > >_There are also no statements on the trade-off between sample fidelity, training difficult and amount of (training or inference time) compute._
> >
> > [As Reviewer 2 raised a similar point, we post the same response in this paragraph]  We now include plots of FID vs training iterations in Figures 5, 6, and 7 for all datasets and include discussion of the same in the Results section. To summarize here, we note that in Figures 5, 6, and 7, the proposed G-equivariant framework converges faster as a function of training iterations (for all datasets except ANHIR) and starts off with better FID values than all competing methods (for all datasets). However,  group-convolutions do require higher compute times than corresponding standard CNNs as detailed in Appendix C2. This computation vs. efficiency tradeoff is an open question in equivariant-networks research and we expect that future efficient implementations of equivariant nets will lower computational costs.
> >
> > ## References
> > [Odena19] Odena, Augustus. "Open questions about generative adversarial networks." _Distill_ 4.4 (2019): e18.
> >
> > [Karras20] Karras, Tero, et al. "Training Generative Adversarial Networks with Limited Data." _arXiv preprint arXiv:2006.06676_ (2020).
> >
> > [Bińkowski18] Bińkowski, Mikołaj, et al. "Demystifying MMD GANs." _International Conference on Learning Representations_. 2018.

---

### Official Review · AnonReviewer1 · 2020-10-28
**Interesting approach that could be enriched.**

**Rating:** 6
**Confidence:** 4

**Review:**

Short summary of the paper:

This paper proposes a new type of architecture that incorporates group-equivarient layers into the discriminator and the generator of a GAN. This group shall be a group of symmetry of the signals to generate, and in this paper, the authors focus on the group generated by translations, pi/2-rotations and reflexions. They also employ linear upsampling, and they show that with no specific ad-hoc tricks(here, class-conditional batch norm & spectral normalisation), it is possible to train competitive GANs with such priors. The experiments are conducted on datasets which have exactly two symmetries (translation or roto-translation), like rotated-MNIST, or approximatively one like Food-101, or CIFAR-10. The major contribution of this paper is experimental, showing better per-class synthesis and claiming one needs less data thanks to those priors.

Pros:
- This is an interesting attempt to incorporate group structure in GANs which can only benefit to the community.
- Various experiments are conducted which positively support the insights of the paper, measured thanks to a visual fidelity metric.

Cons:
- I think the formalism could be more complete and rich. Several points are not addressed, like dealing with more complicated groups that imply an inexact covariance but are still groups of variability. Currently, I find it too elementary.
- From the experimental points of view, I thought small data meant to use at least an order of magnitude less samples than the original dataset for training GANs. At least, it would have been nice to show a more complete plot, in order to understand the tendency is a really limited sample setting, rather than 3 data points for 33%, 66% and 100% of the data.
- I would have appreciated to understand the methodology to pick the hyper parameters. In a supervised context, I understand that one performs a grid search on a validation set which is a subset of the training set. Here, the way to pick an architecture is unclear to me. For instance and in one word, it should be clear if modifying an architecture to make it G-convolutional friendly.

Specific remarks/questions:
- I find quite strange to see no Mallat's citation, whereas the first works on invariance in image processing/CNNs is from his group and since at least 2010 if not older ( https://www.di.ens.fr/data/publications/papers/Eusipco2010InterConfPap.pdf ). Furthermore, I noticed some generative models based on a scattering transform ( https://arxiv.org/abs/1805.06621 / https://arxiv.org/abs/1809.06367 ), yet they do not work substantially better than other architectures whereas the scattering transform employed there leads to no loss of information. It indicates that group a framework might not be the ultimate key to generating images. Consequently, I would substantially lower the claim "to our knowledge, we are the first to introduce group equivariance to GANs, and use geometric considerations in both generator and discriminator".
- This leads to a second remark: the groups of equivariance which are used correspond roughly to horizontal and vertical reflexions. They are not natural groups to consider for image variabilities (like small rotations, or small deformations) Consequently, it is natural that applying a data augmentation strategy based on those groups would "hurt" the performance.
- Again, the group-a priori is quite limited: it's not a generic group which is used, but a very small subset subgroup of the rotation. This is probably due to aliasing but one can wonder how to extend this method to real group of variabilities (deformations, rotation, scaling) This should be discuss, or at least tried, even if it doesn't work.
-  It is written that the authors tried to learn "a transformation from G to Z^2" that could be learned. I think the formalism here is important: it is a representation from G to L^2(Z^2) that one would like to design. I think it would be also nice to develop more this aspect in the paper or not to mention it at all.
- The number of samples used isn't really impressive. 33% of the data isn't order of magnitude smaller than the original dataset thus I find the small data claims strange; what happens if less data is used for instance?
- Why isn't the linear  averaging along the group in the G-equivariant architectures, and why is it only w.r.t. translations? I'm slightly confuse by the combination of the max-pool and global average pool.
- I was wondering if the authors tried to manipulate the orbits of the generated signal: could one use the same "displacement" (or rotation) in the latent space to sample two rotated images from each other with the generator?

Suggestions for improving the paper:
- I believe a significant improvement could be obtain from a more careful formalism to handle complex groups of variability of images.
- I would add slightly more elements about the experimental process, for being more reproducible.
- I think it'd be okay to show some negative results.

[Post-rebuttal] I've read the rebuttal, which answers to many points I raised. I've reflected it in my score.

---

> ### Author Response · Authors · 2020-11-23
> **Reviewer 1 Response (part 1)**
>
> Thank you very much for your time and in-depth review.
>
> >  _This is an interesting attempt to incorporate group structure in GANs which can only benefit to the community. Various experiments are conducted which positively support the insights of the paper_
>
> Thank you for the encouraging assessment of our motivation and design.
>
> > _Significant improvement could be obtained from careful formalism to handle complex groups of variability of images._
>
> We fully agree that handling more complicated groups for general convolutional networks is a very interesting problem. However, to our knowledge, no current _trainable_ group-equivariant CNN framework is simultaneously equivariant to the full spectrum of variabilities seen in natural images (e.g., jointly handling roto-translations, scaling, small deformations parameterized as diffeomorphisms, etc.). Please see Table 1 Page 18 of [Cohen19](https://arxiv.org/pdf/1811.02017.pdf) for a summary of current group-equivariant frameworks. We believe that such a contribution would be of major interest to the community, but would the subject of a separate paper outside of the GAN scope we work in.
>
> Our goal is to show that with careful modification and exhaustive experimentation, modern high-performing class-conditional GAN architectures can be notably improved via equivariance to roto-reflections (8 symmetries). As mentioned in the Discussion section, we could have incorporated a larger set of symmetries via SE(2)-equivariant CNNs for example. However, adding more filter rotations may lead to diminishing returns as seen in Fig. 6 and 7 of [Lafarge20a] and Table 1 of [Graham20]. We therefore limit ourselves to four 90-degree rotations and four reflections and found a substantial increase in GAN performance across a variety of datasets. As noted by Cohen and Welling, [Lenc15] finds that standard classification networks trained on ImageNet do learn invariance to flips and rotations on their own, indicating that these are indeed useful inductive biases for general images.  Based on this feedback, we have expanded our Discussion section to mention limitations w.r.t. arbitrary transformations.
>
> As the reviewer mentioned inexact covariance, GANs can be encouraged to have such properties via balanced consistency regularization (bCR). In our experiments, we find that equivariant networks outperform bCR when the augmentation is restricted to the group considered.
>
> Tangentially, as pointed out by the reviewer, scattering networks offer an elegant solution to jointly handle several image variabilities. However, as scattering networks use fixed non-trainable filter banks and require very specific architectural choices such as absolute value non-linearities, they would require a full restructuring of current GAN architectures and training strategies. Please see our response below for a full contextualization of our approach vs. current generative scattering networks.
>
> > _The number of samples isn't impressive. 33% of the data isn't order of magnitude smaller than the original dataset thus I find the small data claims strange; what happens if less data is used?_
>
> [As Reviewer 3 raised a related topic, we post the same response in this paragraph] This is a miscommunication on our part. To clarify, we claim that the real world datasets we use are themselves in the limited data regime. Class-conditional GANs are typically trained on consisting of hundreds of thousands to millions of samples on datasets such as ImageNet (1M samples) and LSUN (1M+ samples), whereas ANHIR, LYSTO, Food-101, and CIFAR-10 have 28407, 20000, 75747, and 50000 training images, respectively. Our largest dataset of 75K samples (Food-101) has 101 categories and GAN training difficulty is known to increase with higher numbers of classes [Odena19], and CIFAR-10 has recently been shown to be a limited data benchmark for GANs as well in contemporaneous work [Karras20]. As a highlight, we improve the baseline FID by 10+ points for Food-101 in the limited regime using roto-translation equivariant networks. We accordingly rephrase our sample complexity claims in the paper for clarity.
>
> To answer this question specifically: the 33/66/100 analysis was performed on the RotMNIST toy dataset of 12000 samples and not on the real-world datasets. As suggested, we have now added extra RotMNIST experiments with an order of magnitude fewer data. Additionally, in the 2 week discussion period, we trained all GAN methods with the same architectures and hyperparameters on all real-world datasets with an order of magnitude less data and found all models to be largely unstable within just a few thousand training iterations. This is to be expected as the GAN hyperparameters and network sizes are highly dependent on data availability, whereas the settings used were tuned on the full (small-scale) datasets.  For example, parameters tuned for training $256 \times 256$ conditional GANs with 20,000 samples on LYSTO will not work for 2000 samples.

---

> > ### Author Response · Authors · 2020-11-23
> > **Reviewer 1 Response (part 2)**
> >
> > (continued from above)
> >
> > > _I would have appreciated to understand the methodology to pick the hyper parameters. In a supervised context, I understand that one performs a grid search on a validation set which is a subset of the training set._
> >
> > The GAN literature generally does not make use of truly held out test data, rather most papers optimize architectures and hyperparameters to minimize a metric (eg FID) on a validation set. Accordingly, in our revision, we change all occurrences of "test set" to "validation set". To the specific question, we pick our initial hyperparameters and architectures based on the extensive empirical evaluation performed in [Brock19] with further dataset-specific modifications described in Appendix C2. We found optimal hyperparameters and architectures to be sensitive to the dataset and training set up, as is very common with GANs (eg, see page 16 of [Karras19](https://arxiv.org/pdf/1912.04958.pdf)).
> >
> > Briefly, we performed a grid search to identify a stable hyperparameter configuration for all ablations of our method for the ANHIR dataset over learning rates for generator and discriminator, gradient penalty strengths, whether to use batch normalization in the discriminator or not, whether to use average-pooling or max-pooling to reduce spatial extent in the discriminator, and whether to use a Gaussian latent space or a Bernoulli latent space. We use the identified hyperparameter configuration as an initial starting point for all datasets, modifying them as appropriate. We have added these details to Appendix C2 in the revision.
> >
> > > _I find quite strange to see no Mallat's citation / I noticed some generative models based on a scattering transform / I would substantially lower the claim "to our knowledge, we are the first to introduce group equivariance to GANs, and use geometric considerations in both generator and discriminator"._
> >
> > Thank you very much for catching this oversight on our part - we now cite several works on scattering networks.
> >
> > We would like to briefly contrast our work against the two generative scattering papers mentioned:
> > - [Angles18] does not propose a GAN. Instead, they have an encoder-decoder set up where the encoder is a fixed/non-learnable scattering network and the decoder is a standard convolutional network (without equivariance beyond translation). They are roughly similar to variational autoencoders in their design. As shown in Fig. 4 of [Angles18], their approach doesn't yet approach the realism of GANs.
> > - [Oyallon18] do not use scattering networks in either generator or discriminator. Rather, they train a standard GAN (without equivariance beyond translation) to learn the coefficients of a scattering transform based on a training set of precomputed coefficients of images. Once their generator is trained to generate scattering coefficients, they use a scattering transform to construct images.
> >
> > Therefore, we believe that our claim is accurate in spirit and we rephrase our claim in the paper to make it more specific and hopefully clarify any misunderstandings.
> >
> > > _The groups of equivariance which are used correspond roughly to horizontal and vertical reflexions. They are not natural groups to consider for image variabilities (like small rotations, or small deformations) Consequently, it is natural that applying a data augmentation strategy based on those groups would "hurt" the performance._
> >
> > We believe that there may have been a miscommunication. Using those symmetries for augmentation correctly in the GAN context via methods such as balanced consistency regularization (bCR) and auxiliary rotations in fact improves synthesis and does not hurt performance as shown in Table 3. Table 3 further finds that equivariant networks show improvements over these augmentation based approaches when the augmentation is limited to those transformations. However, as noted in the text, using other augmentations (e.g. small diffeomorphic warps) via bCR would benefit all methods whether they be standard translation-equivariant GANs or our rotation+reflection+translation-equivariant GANs. To clarify a minor point, the group used corresponds to 90-degree rotations and mirror reflections (horizontal, vertical, diagonal, anti-diagonal) and not just horizontal and vertical reflections.
> >
> > > _It is written that the authors tried to learn "a transformation from G to Z^2" that could be learned. I think the formalism here is important: it is a representation from G to L^2(Z^2) that one would like to design. I think it would be also nice to develop more this aspect in the paper or not to mention it at all._
> >
> > This sentence has accordingly been removed, thank you for the suggestion.

---

> > > ### Author Response · Authors · 2020-11-23
> > > **Reviewer 1 Response (Part 3)**
> > >
> > > (continued from above)
> > >
> > > > _Why isn't the linear averaging along the group in the G-equivariant architectures, and why is it only w.r.t. translations? I'm slightly confuse by the combination of the max-pool and global average pool._
> > >
> > > To clarify, there are three forms of pooling occurring in our architectures:
> > > 1. We pool along the group/over orientations at each spatial location of the feature map at the output of our architectures to recover a function with domain $Z^2$. For example, in the generator, this final group pooling is used to produce a planar image.
> > > 2.  Standard spatial max-pooling (an equivariant operation) is used in the discriminator to reduce the spatial extent of the feature maps defined on the group as the discriminator is a classification network mapping an input image to a scalar.
> > > 3. Global Average Pooling (an invariant operation) is used in the discriminator after pooling along the group to convert $k \times k \times n$ planar feature maps into an $n$-d vector used for conditioning the discriminator via the projection method of [Miyato18]. If we were working with unconditional GANs, we could use global pooling to map the $k \times k \times n$ planar feature maps into a scalar indicating whether the image is real or synthesized.
> > >
> > > > _I was wondering if the authors tried to manipulate the orbits of the generated signal: could one use the same "displacement" (or rotation) in the latent space to sample two rotated images from each other with the generator?_
> > >
> > > We note that our generator network is not end-to-end equivariant as it uses a linear projection and a reshaping to embed a 128D Gaussian noise vector into a spatial feature map. As the latent space is not disentangled between shape, texture, and pose, we will not see such predictable behavior. However, contemporaneous work in [Lafarge20] which we mention in the Discussion section proposes an SE(2)-equivariant variational autoencoder with separate structured latent spaces, which may be incorporated into our framework in future work to enable explicit control.
> > >
> > > > _I would add slightly more elements about the experimental process, for being more reproducible._
> > >
> > > We have now added further experimental details to Appendix C2.
> > >
> > > > _I think it'd be okay to show some negative results._
> > >
> > > We believe that the paper does show a few negative results.
> > > 1. We show several collages of random samples for all datasets in Appendix A, some of which are not realistic.
> > > 2. For CIFAR-10, using G-CNNs in the generator with a standard CNN discriminator slightly worsens synthesis (as measured by FID). This may be due to suboptimal hyperparameters for that specific setting as the same hyperparameters were used for all experiments for a given dataset.
> > >
> > > We would be happy to include more negative results in the final version if we misunderstood the experimental aspect you may be referencing.
> > >
> > > ## References
> > > [Oyallon18] Scattering networks for hybrid representation learning. TPAMI 2018.
> > >
> > > [Angles18] Generative networks as inverse problems with Scattering transforms. ICLR 2018.
> > >
> > > [Lafarge20a] Roto-translation equivariant convolutional networks: Application to histopathology image analysis. arXiv 2020.
> > >
> > > [Lafarge20b] Orientation-Disentangled Unsupervised Representation Learning for Computational Pathology. 2020.
> > >
> > > [Graham20] Dense Steerable Filter CNNs for Exploiting Rotational Symmetry in Histology Images. arXiv 2020.
> > >
> > > [Miyato18] cGANs with Projection Discriminator. ICLR 2018.
> > >
> > > [Brock19] Large Scale GAN Training for High Fidelity Natural Image Synthesis. ICLR 2018.
> > >
> > > [Lenc15] Understanding Image Representations by Measuring Their Equivariance and Equivalence. CVPR 2015.

---

### Official Review · AnonReviewer2 · 2020-10-28
**Needs a little more**

**Rating:** 5
**Confidence:** 2

**Review:**

This paper presents a method for incorporating inductive symmetry priors into the network architectures of GANs. The authors propose replacing standard CNNs with group-equivariant CNNs in either the generator or discriminator or both. The method is evaluated by comparing generated images to originals using Frechet distance for a number of datasets.

The paper is well written and comprehensive. Its main contribution seems to be the adaptation of some common GAN techniques to the group-equivariant case.

The experiments are fairly convincing, although I would like to see more discussion about when to use the group convolution in the discriminator vs generator vs both, as this seems to make a difference for different datasets. I am not an expert and am unsure if Frechet distance is the right metric here, but it seems to be common at least.

The authors say they compare their method to one of the two main, state-of-the-art GAN designs. It would be interesting to see a comparison with other flavours of GAN also. Given the empirical nature of the paper, I think this would be a more convincing argument (i.e. group-convolutions can improve GANs not just BigGAN).  It would also be interesting to see how the different methods perform given different amounts of training time.

In general I like this paper, though I think the experiments could be more comprehensive (as noted above). That being said, it is not immediately clear to me that this is enough of a contribution to be accepted, since it is just applying existing methods in the group-equivariant case. As I've said though, I am not an expert in GANs, and I would be open to increasing my rating if more knowledgeable reviewers disagree on this point.

---

> ### Author Response · Authors · 2020-11-23
> **Reviewer 2 Response (part 1)**
>
> We thank the reviewer for their valuable analysis and experimental suggestions. To address individual points:
>
> > _The paper is well written and comprehensive. Its main contribution seems to be the adaptation of some common GAN techniques to the group-equivariant case._
>
> Thank you for the positive assessment of the paper's writing and thoroughness. We would like to highlight that alongside incorporating group-equivariance within generative adversarial models, we show that group-equivariant networks can be used to enhance modern high-performing class-conditional GANs in the limited data regime, can outperform some GAN-specific augmentation based approaches (when the augmentation is restricted to the same group), and even benefit synthesis when the images have preferred orientation (as in Food-101 and CIFAR-10).
>
> Further, based on your suggestions, we now show that G-equivariance is generically beneficial to GANs, please see below.
>
> > _It would be interesting to see a comparison with other flavours of GAN. Given the empirical nature of the paper, I think this would be a more convincing argument_
>
> Thank you for this suggestion. To show the benefits of group-equivariance to other varieties of GANs, we now include experiments on GANs for image-to-image translation (Google Maps to Aerial Photos) in App. B and find that $p4$-equivariance improves translation performance over the optimized baseline by ~100%. Also, we note that the Rotated MNIST experiments were performed with straightforward architectures due to the relative simplicity of the dataset, so they can be considered to be in the family of DCGAN approaches and not BigGAN.
>
> > _The experiments are fairly convincing, although I would like to see more discussion about when to use the group convolution in the discriminator vs generator vs both_
>
> This phenomenon was originally referenced in Section 3.2/Results and we fully agree that it deserves more discussion. Due to page limits, we briefly expanded our discussion of this dataset sensitivity and will make sure to include more details in the final version of the paper.
>
> > _I am not an expert and am unsure if Frechet distance is the right metric here, but it seems to be common at least._
>
> As stated by the reviewer, generative visual synthesis evaluation is typically performed via Frechet distances. To elaborate, the Frechet Inception Distance (FID) is the 2-Wasserstein distance between real and fake data distributions (fitted via multivariate Gaussians) in the feature space of a pre-trained network. FID has been found to correlate well with human perception [Heusel17, Fig.3] and is universally used for GAN evaluation. However, FID does have associated issues and we accordingly expand our discussion of the same in Section 3. To provide a more holistic evaluation, we also provided precision and recall scores for GANs [Kynkäänniemi19] in Figure 5, finding that group-equivariance does typically help GAN training.
>
> > _It would be interesting to see how the different methods perform given different amounts of training time._
>
> [As Reviewer 3 raised a similar point, we post the same response in this paragraph] We now include plots of FID vs training iterations in Figures 5, 6, and 7 for all datasets and include discussion of the same in the Results section. To summarize here, we note that in Figures 5, 6, and 7, the proposed G-equivariant framework converges faster as a function of training iterations (for all datasets except ANHIR) and starts off with better FID values than all competing methods (for all datasets). However,  group-convolutions do require higher compute times than corresponding standard CNNs as detailed in Appendix C2. This computation vs. efficiency tradeoff is an open question in equivariant-networks research and we expect that future efficient implementations of equivariant nets will lower computational costs.

---

> > ### Author Response · Authors · 2020-11-23
> > **Reviewer 2 Response (part 2)**
> >
> > (continued from above)
> >
> > > _In general I like this paper, though I think the experiments could be more comprehensive (as noted above). That being said, it is not immediately clear to me that this is enough of a contribution to be accepted, since it is just applying existing methods in the group-equivariant case._
> >
> > [As Reviewer 4 made a similar point, we post the same response in this paragraph] We would like to emphasize that this is not the first time that equivariant nets have been used for generative adversarial visual models with previous attempts using capsule networks [Jaiswal18] or (as pointed out by Reviewer 1) scattering networks [Oyallon18]. None of these approaches so far have been shown to clearly outperform baseline GANs, whereas our proposed framework does, scaling to images with a high number of classes or resolution. In addition, it outperforms recent methods which try to approximate equivariance in GANs via data augmentation. We also now show experiments on image-to-image translation (as suggested by Rev. 2), that show that equivariant networks generically benefit GAN tasks. We believe that once it is convincingly shown that equivariant networks enhance GAN training, future theoretical and further empirical investigations will follow on the interplay between equivariance and GANs.
> >
> >
> > ## References
> > [Heusel17] Heusel, Martin, et al. "Gans trained by a two time-scale update rule converge to a local nash equilibrium." _Advances in neural information processing systems_. 2017.
> >
> > [Kynkäänniemi19] Kynkäänniemi, Tuomas, et al. "Improved precision and recall metric for assessing generative models." _Advances in Neural Information Processing Systems_. 2019.
> >
> > [Jaiswal18] CapsuleGAN: Generative Adversarial Capsule Network, *ECCV Workshops.* 2018.
> >
> > [Oyallon18] Oyallon, Edouard, et al. "Scattering networks for hybrid representation learning." _IEEE transactions on pattern analysis and machine intelligence_ 2018.

---

### Official Review · AnonReviewer4 · 2020-10-28
**Review of ICLR Paper 1928: Written well but needs some improvements**

**Rating:** 6
**Confidence:** 4

**Review:**

# Review of ICLR Paper 1928

## Group Equivariant Generative Adversarial Networks

This manuscript addresses the problem of artificially generating images for which the label should be unaffected by certain symmetries and/or translations. To address this problem, the authors use the Group Equivariant Convolutional Neural Networks of Cohen \& Welling (2016) as the generator and/or discriminator in a Generative Adversarial Network. They conduct a sequence of experiments on smaller data sets--four real and one synthetic--exhibiting varying levels of symmetry, which together suggest that Group Equivariant GANs may be more effective than traditional convolutional networks in the low data setting.

My assessment of this manuscript is that it sits right on the threshold separating acceptance and rejection. On one hand, the writing is remarkably clear of grammatical errors and the authors are evidently deeply familiar with GAN literature. On the other hand, the paper's novelty is minimal and the mathematical descriptions of some important parts of the manuscript are insufficient. Group Equivariant Networks were introduced in 2016, and in the original paper those authors demonstrated the efficacy and training efficiency of these networks. In this manuscript, the authors' contribution can be summarized as placing this architecture in a GAN and observing similar improvements, which is unsurprising and in my assessment a marginal contribution.

I've summarized further concerns below.

- Section 2.1, the Preliminaries section, makes the fundamental concepts extremely hard to understand.
    - "Equivariant" and "invariant" are not defined.
    - Please provide a citation for the plane symmetry groups $p4$ and $p4m$.
    - The phrase "an image is a function $f$ on a rectangular grid $\mathbb{Z}^{2}$" is ambiguous. Use the notation $f: \mathbb{Z}^{2} \to \mathbb{R}$. However, you later mention that $f$ has $K$ channels, in which case $f: [K] \times \mathbb{Z} \times \mathbb{Z} \to \mathbb{R}$. Regardless, I'm confused.
    - Which operation defines your group? You have sums and product and compositions, so this should be clarified.
    - Define a filter $\psi$.
    - The $g^{-1} y = y - g$ on page 3 is confusing. How can this work when $g \in p4$ is a rotation and not a translation?
    - On the whole, I find this section inadequate. The Cohen paper introduces these concepts well, so I encourage you to stick closer to the mathematics introduced there.
- "We finally max-pool over the set of transformations to obtain the generated image $x \in \mathbb{Z}^{2}$". An image is a function, not a pair of integers.
- "The group-equivariant discriminator receives an inputs $x \in \mathbb{Z}^{2}$" Again, an image is not a pair of integers.
- Table two is problematic for me. Your Group-Equivariant networks have less parameters, and you are operating in a self-professed limited data environment. Shouldn't I expect that the networks with less parameters exhibit less overtraining?
- What are the details of how you conducted the ablation study? What layers and/or features did you remove?
- I am not convinced that, in table 2, the potential for augmentation leaking means that the CNN with standard augmentation is "inapplicable". Why don't you just let the FID evaluation measure its performance, which should be higher if the network has captured symmetries incorrectly?

---

> ### Author Response · Authors · 2020-11-23
> **Reviewer 4 Response**
>
> Thank you very much for the in-depth reading, corrections, and relevant comments and questions.
>
> > _In this manuscript, the authors' contribution can be summarized as placing this architecture in a GAN and observing similar improvements, which is unsurprising and in my assessment a marginal contribution._
>
> While we agree that in retrospect, group-equivariant networks benefiting GAN frameworks may not be surprising, we would like to emphasize that this is not the first time that equivariant nets have been used for generative adversarial visual models with previous attempts using capsule networks [Jaiswal18] or (as pointed out by Reviewer 1) scattering networks [Oyallon18]. None of these approaches so far have been shown to clearly outperform baseline GANs, whereas our proposed framework does, scaling to images with a high number of classes or resolution. In addition, it outperforms recent methods which try to approximate equivariance in GANs via data augmentation. We also now show experiments on image-to-image translation (as suggested by Rev. 2), that show that equivariant networks generically benefit GAN tasks. We believe that once it is convincingly shown that equivariant networks enhance GAN training, future theoretical and further empirical investigations will follow on the interplay between equivariance and GANs. [As Reviewer 2 made a similar point, we posted the same response in this paragraph]
>
> > _Section 2.1, the Preliminaries section, makes the fundamental concepts extremely hard to understand._
>
> Thank you for the careful read and suggestions. We have extensively revised the presentation of fundamental background concepts in the updated submission.
>
> > _"We finally max-pool over the set of transformations to obtain the generated image  x∈Z2". An image is a function, not a pair of integers._
> > _"The group-equivariant discriminator receives an inputs  x∈Z2" Again, an image is not a pair of integers._
>
> Thank you for catching these typos, they have been fixed.
>
> > _Table two is problematic for me. Your Group-Equivariant networks have less parameters, and you are operating in a self-professed limited data environment. Shouldn't I expect that the networks with less parameters exhibit less overtraining?_
>
> We believe that there may have been a couple of misunderstandings. To clarify,
> - We use fewer group filters, however, all networks (group equivariant or standard) for a given dataset have roughly the same number of parameters. This was stated in paragraph 1 of the experiments section as: "For each comparison, the number of group-filters in each layer is divided by the square root of the cardinality of the symmetry set to ensure a similar number of parameters to the standard CNNs to enable fair comparison". A very small difference in parameter count comes from the $p4$ (or $p4m$) batch normalization layers requiring fewer affine scale and shift parameters as they are per each *group* feature map (shared among orientations) rather than each standard feature map. This follows experimental convention in the equivariant learning literature.
> - More importantly, the referenced table (now Table 3) presents FID values where the lower values we achieve indicate less overtraining/improved synthesis and generalization to a held out set. In all four datasets, replacing both the generator and discriminator (or discriminator alone) with equivariant networks lowers FID scores and thus improves synthesis. Further, in three out of four datasets, using an equivariant network as generator improves over standard GAN training as well.
>
> We will endeavor to make this clearer in the final version of the paper.
>
> > _What are the details of how you conducted the ablation study? What layers and/or features did you remove?_
>
> We have now expanded our description of the ablation study in Appendix C2.
>
> > _I am not convinced that, in table 2, the potential for augmentation leaking means that the CNN with standard augmentation is "inapplicable". Why don't you just let the FID evaluation measure its performance, which should be higher if the network has captured symmetries incorrectly?_
>
> We agree that this claim should be demonstrated. We now include FID numbers for training GANs with standard augmentation in the referenced table. As expected, they are significantly worse than training without augmentation due to the transformations leaking into the generated images.
>
> ## References
> [Jaiswal18] CapsuleGAN: Generative Adversarial Capsule Network, *ECCV Workshops.* 2018.
>
> [Oyallon18] Oyallon, Edouard, et al. "Scattering networks for hybrid representation learning." _IEEE transactions on pattern analysis and machine intelligence_ 2018.

---

### Author Response · Authors · 2020-11-23
**Overall Response**

We are grateful to the reviewers for their time, expertise, and constructive comments. We have carefully incorporated their feedback to considerably improve the revised submission.

We are happy to see that the reviewers found our initial submission interesting (R1 and R3), beneficial to the community (R1), and well-written (R2 and R4), with comprehensive (R1 and R3) and convincing (R2) experimentation. Some common concerns raised were regarding the significance of the contribution, lack of description of training tradeoffs, and mathematical formalism. These concerns (and all others) are addressed in detail in individual review responses below and we hope that they are satisfactorily resolved.

To summarize changes in the updated submission:
- As suggested by Reviewer 2, we now show that roto-translation equivariance benefits GAN tasks generically (and not just in the context of our original submission) via a pilot study on image-to-image translation in Appendix B. Briefly, the $p4$-equivariant translation GAN doubles the performance of an optimized standard translation GAN.
- We have rewritten Section 2.1 to improve its mathematical clarity and better present foundational ideas, based on feedback from Reviewer 4.
- As suggested by Reviewer 4, we add experiments for Food-101 and CIFAR-10 synthesis using standard augmentation techniques in the GAN context in Table 3 (Table 2 in the original submission).
- We have added a 10% data experiment to Table 2 (Table 1 in original submission) as suggested by Reviewer 1. As the 2-week discussion period allowed for extra experimentation, we repeat the Rotated MNIST generation experiments with multiple random seeds and present new results for all data quantity settings and ablations.
- To address questions regarding training tradeoffs, we have added convergence plots (Figure 5 in the main text; Figures 6 and 7 in Appendix A) and discuss computational costs in Appendix C2. In the revision, we removed the boxplots in the appendix present in the original submission as Figure 6 covers similar ground.
- We now include the related work on Scattering networks pointed out by Reviewer 1.
- We expand the relevant parts of Sections 3 and 4 to mention limitations regarding evaluation metrics of GANs (raised by R3) and choice of group (raised by R1).
- We moved the dataset description table from the appendix to the main text to better emphasize the fact that the real-world datasets we experiment on are limited data benchmarks.

To reiterate, we deeply appreciate the feedback and are happy to make any further changes suggested by the reviewers.

---

### Decision · Program_Chairs · 2021-01-07
**Final Decision**

**Decision:**

Accept (Poster)

**Comment:**

This paper use Group convolutional neural networks in both generators and discriminator of GANs, and demonstrates advantages of this approach when training with a relatively small sample size. While the novelty is limited in the work  as it simply applies G-CNN for GANs , I believe this application is interesting and  the authors have applied it to many GAN image synthesis applications (conditional generation , pix2pix) on various benchmarks, which gives  evidence of  the potential of GCNNs in generative modeling. Accept